# Test like you Train in Implicit Deep Learning

## Abstract

Implicit deep learning has recently gained popularity with applications ranging from meta-learning to Deep Equilibrium Networks (DEQs). In its general formulation, it relies on expressing some components of deep learning pipelines implicitly, typically via a root equation called the inner problem. In practice, the solution of the inner problem is approximated during training with an iterative procedure, usually with a fixed number of inner iterations. During inference, the inner problem needs to be solved with new data. A popular belief is that increasing the number of inner iterations compared to the one used during training yields better performance. In this paper, we question such an assumption and provide a detailed theoretical analysis in a simple affine setting. We demonstrate that overparametrization plays a key role: increasing the number of iterations at test time cannot improve performance for overparametrized networks. We validate our theory on an array of implicit deep-learning problems. DEQs, which are typically overparametrized, do not benefit from increasing the number of iterations at inference while meta-learning, which is typically not overparametrized, benefits from it.

## 1 Introduction

Implicit deep learning has seen a surge in recent years with various instances such as Deep Equilibrium Networks (DEQs; Bai et al. 2019), OptNets (Amos & Kolter, 2017), Neural ODEs (Chen et al., 2018), or meta-learning (Finn et al., 2017; Rajeswaran et al., 2019). Here, we define implicit deep learning as the setting where intermediary outputs or task-adapted parameters $z^\star \in \mathbb{R}^{d_z}$ are defined *implicitly* as the solution of a root-finding problem involving the parameters $\theta \in \mathbb{R}^{d_\theta}$ and the training set $\mathcal{D}_{\text{train}}$, that is, $f(z^\star, \theta, \mathcal{D}_{\text{train}}) = 0$. Changing $\theta$ or $\mathcal{D}_{\text{train}}$ changes the landscape of $f$, thus $z^\star$ depends on $\theta$ and $\mathcal{D}_{\text{train}}$. The optimal parameters $\theta$ are found by minimizing a loss function $\ell(z^\star, \mathcal{D}_{\text{train}})$, which depends on $\theta$ implicitly through $z^\star$. Hence, the learning problem has the following *bilevel* structure:

$$\arg\min_\theta \ell(z^\star(\theta, \mathcal{D}_{\text{train}}), \mathcal{D}_{\text{train}}) \quad \text{s.t.} \quad f(z^\star(\theta, \mathcal{D}_{\text{train}}), \theta, \mathcal{D}_{\text{train}}) = 0. \tag{1}$$

We explain in detail how this formulation covers different cases such as Implicit Meta-learning (iMAML; Rajeswaran et al. 2019) and Deep Equilibrium Models (DEQs; Bai et al. 2019) in Section 2. The gradient of the loss $\ell$ relative to $\theta$ –called *hypergradient*– can be computed using the implicit function theorem (IFT; Krantz & Parks 2013; Blondel et al. 2022), and then used to learn the optimal $\theta^\star$ by gradient descent. Once the optimal $\theta^\star$ is learned, the model is used at inference with a new dataset $\mathcal{D}_{\text{test}}$, and we need to find a new root $z^\star(\theta^\star, \mathcal{D}_{\text{test}})$.

In the ideal formulation of (1), the output of the model is independent of the root-finding procedure used to solve the inner problem. In other words, there is "a decoupling between representational capacity and inference-time efficiency" as explained by Bai (2022) for DEQs.

However, in most cases $z^\star(\theta)$ can only be approximated using an iterative procedure –note that for conciseness, we omit $\mathcal{D}_{\text{train}}$ in quantities that depend on it, but remain explicit for dependency on other datasets. For example, DEQs rely on Broyden's method (Broyden, 1965) or Anderson Acceleration (Anderson, 1965), while meta-learning uses gradient-based methods. Although some works have tried to tackle the question of how to speed up this iterative procedure by using a learned procedure (Bai et al., 2022b), warm starting (Micaelli et al., 2023; Bai et al., 2022a), or accelerating the hypergradient computation (Fung et al., 2022; Ramzi et al., 2022), in most cases, the number of iterations $N$ used for this procedure is fixed during training in order to keep a reasonable computational budget.[1] We denote the resulting approximation $z_N(\theta)$. The practical problem solved

---

[1] See the original implementations for DEQs or implicit meta-learning.

in implicit deep learning then becomes:

$$\theta^{\star,N} \in \arg\min_{\theta} \ell(z_N(\theta)) \text{ s.t. } z_N(\theta) \text{ is the } N\text{-th iterate of a procedure solving } f(z,\theta) = 0. \quad \text{(P)}$$

Here, we highlight the dependency of the solution $\theta^{\star,N}$ on the number of inner iterations $N$. At inference, one should decide how many iterations are used to solve the inner problem. We ask:

---
**Question**

Is there a performance benefit in changing the number of inner iterations $N$ once the model is fitted for implicit deep learning?

---

Because of the decoupling in the ideal case, many papers hypothesize that, when the model is fixed, a better approximation of the solution of the inner problem –i.e. increasing $N$– could bring performance benefits (Bai, 2022; Pal et al., 2022). For instance, Gilton et al. (2021) state that "the computational budget can be selected at test time to optimize context-dependent trade-offs between accuracy and computation". However, they only show that performance deteriorates when the number of test-time inner iterations is lower than in training. They also suggest that networks trained with IFT are more robust to changes in the number of inner iterations than their unrolled counterparts, where the hypergradient is computed via backpropagation through the inner solver steps. Anil et al. (2022) ask whether more test-time iterations can help tackle "harder" problems – for example increasing the dimension in a maze resolution problem. They empirically show that if a DEQ has a property termed path-independence, it can benefit from more test-time iterations to improve the performance out-of-distribution. However, there is little evidence that increasing the number of iterations at test-time can improve the performance for DEQs on the same data distribution. Conversely, in the meta-learning setup, the number of inner iterations can be greater during inference than during training, for example in the text-to-speech experiment of Chen et al. (2020) –100 iterations during training vs 10 000 iterations at test-time. Finn et al. (2017) also showed that the performance increases when the number of inner iterations increases at test-time.

Formally, we would like to know if changing the number of iterations to approximate the root from $N$ to $N + \Delta N$ with $\Delta N > -N$ –i.e., using $z_{N+\Delta N}(\theta^{\star,N})$ instead of $z_N(\theta^{\star,N})$– can yield better performance. In this paper, we answer that question first with a theoretical analysis in a simple affine case and then with extensive empirical results. In our theoretical analysis, we study $D(N, \Delta N) \overset{\text{def}}{=} \ell(z_{N+\Delta N}(\theta^{\star,N})) - \ell(z_N(\theta^{\star,N}))$, the training loss increase when changing the number of inner iterations by $\Delta N$ for a fixed learned $\theta^{\star,N}$. This quantity is a proxy for the increase in test loss, provided we have access to enough training data. On the other hand, we consider changes in test-set metrics in our experiments.

Theoretically, we uncover for overparametrized models a phenomenon we term **Inner Iterations Overfitting (I2O)**, in which there is no benefit in changing the number of inner iterations. We empirically find that DEQs suffer from I2O, while we confirm the benefit of increasing the number of inner iterations for the meta-learning setup. Our contributions are the following:

- The **theoretical demonstration of I2O** for the case where $f$ is affine. In Theorem 1, we derive a lower bound on $D(N, \Delta N)$ and characterize two regimes in subsequent corollaries: overparametrization in $\theta$ which leads to I2O and no overparametrization. We also characterize the implications of this result for non-affine $f$ in Theorem I.10, and for the generalization setting in Corollary D.1. This provides a practical guideline for DEQs that fall close to the overparametrization regime: in order to achieve the best performance at test time, one should use the same number of inner iterations as in training.

- The **empirical demonstration of I2O** for DEQs on diverse tasks such as image classification, image segmentation, natural language modeling, and optical flow estimation. This validates the practical guideline established above in a generalization setting, and also the current practice. We also show that I2O is much less prevalent in meta-learning cases.

- The **comparison of robustness to changes in inner iterations number** between IFT and unrolling. Theorem 2 shows that the choice of hypergradient computation does not impact whether implicit deep learning suffers from I2O or not. We highlight this phenomenon empirically for DEQs.

## 2 BACKGROUND ON IMPLICIT DEEP LEARNING

**DEQs** DEQs were introduced by Bai et al. (2019) for NLP and have since then been used for a variety of tasks including computer vision (Bai et al., 2020; Micaelli et al., 2023), inverse problems (Gilton et al., 2021; Zou et al., 2023) or optical flow estimation (Bai et al., 2022a). On optical flow estimation (Bai et al., 2022a) and landmark detection (Micaelli et al., 2023), they have even managed to achieve a new state of the art. In their ideal formulation, DEQs are trained by minimizing a task-specific loss on a dataset, where the output of the model is obtained by computing the fixed point of a nonlinear function (Bai et al., 2019):

$$\arg \min_{\theta} \sum_i \ell(z_i^\star(x_i, \theta), y_i) \quad \text{s.t.} \quad z_i^\star(x_i, \theta) = g(z_i^\star(x_i, \theta), x_i, \theta) \tag{2}$$

They can be fitted in the framework introduced in Eq. (1) by lifting the inner variables $z_i$ to a stacked version $\mathbf{z} = [z_1, \ldots, z_n]^\top$ and similarly for the inner functions and the outer losses. The inner problem can also be cast as a root finding problem rather than a fixed point problem, by simply considering $f(z, x_i, \theta) = z - g(z, x_i, \theta)$. The gradient of the outer losses with respect to the parameters $\theta$ is then computed using the IFT, which yields:

$$\nabla L_i = - \left( \partial_z f\left(z^\star, x_i, \theta\right)^{-1} \partial_\theta f\left(z^\star, x_i, \theta\right) \right)^\top \nabla_z \ell\left(z^\star, y_i\right), \tag{3}$$

with $L_i(\theta) = \ell(z_i^\star(x_i, \theta), y_i)$. Importantly, this gradient computation does not depend on the procedure used to compute $z^\star$. Therefore, the intermediary outputs (activations) used to compute it do not need to be saved in order to compute the gradient, leading to a memory-efficient scheme.

**(i)MAML** Model-agnostic Meta-learning (MAML) was introduced by Finn et al. (2017) as a technique to train neural networks that can quickly adapt to new tasks. The idea is to learn a meta-model, such that its parameters when trained on a new task yield good generalization performance. Formally, MAML has the following formulation:

$$\arg \min_{\theta^{(\text{meta})}} \sum_i \ell\left(\theta_i, \mathcal{X}_i^{(\text{val})}\right) \quad \text{s.t.} \quad \theta_i = \theta^{(\text{meta})} - \alpha \nabla_\theta \ell(\theta^{(\text{meta})}, \mathcal{X}_i^{(\text{train})}), \tag{4}$$

where we omitted the dependence of $\theta_i$ on $\theta^{(\text{meta})}$ and $\mathcal{X}_i^{(\text{train})}$ for conciseness. The right-hand-side corresponds to the training on a new task: a single gradient descent step with step size $\alpha$ to minimize a task-specific training loss, the task $i$ being defined by its training and validation datasets $\mathcal{X}_i^{(\text{train})}$ and $\mathcal{X}_i^{(\text{val})}$. The task-adapted parameters $\theta_i$ are then used to compute the loss on the validation set to measure how well these parameters generalize. In follow-up work, Rajeswaran et al. (2019) introduced the implicit MAML (iMAML) formulation, where the gradient descent step is replaced by the minimization of a regularized task-specific loss. Formally, the problem is:

$$\arg \min_{\theta^{(\text{meta})}} \sum_i \ell\left(\theta_i, \mathcal{X}_i^{(\text{val})}\right) \quad \text{s.t.} \quad \theta_i \in \arg \min_{\theta} \underbrace{\ell(\theta, \mathcal{X}_i^{(\text{train})}) + \frac{\lambda}{2} \|\theta - \theta^{(\text{meta})}\|_2^2}_{F(\theta, \theta^{(\text{meta})})} \tag{5}$$

This formulation can easily be cast into the form of Eq. (1) by replacing the inner optimization problem with the associated root problem on the gradient $\nabla_\theta F(\theta, \theta^{(\text{meta})}) = 0$. Then, similarly to the DEQ setting, the task-adapted parameters $\theta_i$ can be stacked together to form a single inner variable $\mathbf{z} = [\theta_1, \ldots, \theta_n]^\top$.

## 3 THEORY OF AFFINE IMPLICIT DEEP LEARNING

In order to study the I2O phenomenon, we will make some simplifying assumptions on the nature of the inner procedure of problem (P) as well as the functions studied. First, we consider that the procedure to solve the inner problem is a fixed-point iteration method with fixed step sizes. Second, we restrict ourselves to the case where the inner problem is affine. Third, we consider only cases where the outer loss is quadratic.

Before we proceed to the formal demonstration of I2O, let us give some intuition. When the inner problem is overparametrized in the outer variable, it means that we can tune the approximate output of the procedure $z_N$ to minimize exactly the outer loss. But this tuning is highly dependent on the procedure, and therefore on the number of inner iterations $N$ used. If it is changed while keeping the same learned outer variable, the outer loss cannot decrease because it is already minimized.

**Main result and corollaries** Let us formalize the assumptions used to prove our main result.

**Assumption 3.1** (Fixed-point iteration). *The procedure to solve the inner problem $f(z, \theta) = 0$ is a fixed-point iteration method with fixed step size $\eta > 0$ and initialization $z_0 \in \mathbb{R}^{d_z}$. This means that:*

$$z_{N+1}(\theta) = z_N(\theta) - \eta f(z_N(\theta), \theta) \ . \tag{6}$$

For iMAML, this corresponds to gradient descent to solve the task adaptation, with $f = \nabla_z F$ with $F$ the regularized task-specific loss. Note that it does not perfectly match the practice for DEQs which are usually trained with Broyden's method (Broyden, 1965) or Anderson acceleration (Anderson, 1965). Also, since $z_0$ does not depend on $\theta$, this does not cover the MAML setting.

**Assumption 3.2** (Affine inner problem). *$f : \mathbb{R}^{d_z \times d_\theta} \to \mathbb{R}^{d_z}$ is an affine function:*

$$f(z, \theta) = K_{\text{in}}^\top (Bz + U\theta + c), \tag{7}$$

*with $K_{\text{in}}, B \in \mathbb{R}^{d_x \times d_z}, U \in \mathbb{R}^{d_x \times d_\theta}, c \in \mathbb{R}^{d_x}$, with $K_{\text{in}}$ surjective, i.e. $d_x \leq d_\theta$. Moreover, $BK_{\text{in}}^\top$ has eigenvalues with positive real part.*

Assumption 3.2 corresponds to considering a DEQ with an affine layer, i.e. for each sample $x_i$ we would have $f(z, \theta, x_i) = K_{\text{in}}(x_i)^\top (B(x_i)z + U(x_i)\theta + c(x_i))$ which is then stacked for the whole dataset $\mathcal{D}_{\text{train}}$ (more on this in Appendix B). This setting is commonly used to study DEQs (Kawaguchi, 2021), although in practice $f$ is nonlinear. Note however that it is different from the parametrization of Winston & Kolter (2020) (discussed in Appendix B). This class of function corresponds to affine functions for which the fixed point iterations (6) converge (see Appendix B). In particular, it is more general than the simple case of $K_{\text{in}} = I$ with $B$ whose eigenvalues have a positive real part. For iMAML, this corresponds to meta-learning a linear model with a quadratic regression loss. We show these two correspondences in Appendix B. More generally, it includes cases where $f$ is the gradient of a convex lower-bounded quadratic function $F(z, \theta) = \frac{1}{2}\|K_{\text{in}}z + U\theta + c\|_2^2$, with $B = K_{\text{in}}$, a setting studied by Vicol et al. (2022). We provide an extended review of this work in Appendix E. While this affine assumption departs from practice, we show in Theorem I.10 (in Appendix I) that if one wants to consider a non-affine function $f$ as is customary in practical implicit models, it is possible to use its Taylor expansion close to the optimum values $(\theta^\star, z^\star(\theta^\star))$ to fit in in this assumption. In this case, the difference between the solutions of (P) for $f$ and its linearized version is controlled by the difference between the initialization $z_0$ and $z^\star(\theta^\star)$.

**Assumption 3.3** (Quadratic outer loss). *$\ell$ is a convex quadratic function bounded from below:*

$$\ell(z) = \frac{1}{2}\|K_{\text{out}}z - \omega\|_2^2, \tag{8}$$

*with $\omega \in \mathbb{R}^{d_\omega}$ and $K_{\text{out}} \in \mathbb{R}^{d_z \times d_\omega}$.*

Assumption 3.3 is meaningful, typically in inverse problems (Zou et al., 2023) or meta-learning regression (Finn et al., 2017; Rajeswaran et al., 2019) where the output of the inner problem is compared to a ground truth signal. It does not include different types of problems such as image classification (Bai et al., 2020) that would require more complex losses such as cross-entropy.

Before stating our main result, we recall that our objective is to control $D(N, \Delta N) \stackrel{\text{def}}{=} \ell(z_{N+\Delta N}(\theta^{\star,N})) - \ell(z_N(\theta^{\star,N}))$, the loss increase when the number of inner iterations changes by $\Delta N$.

---

**Main result**

**Theorem 1** (Inner Iterations Overfitting for affine inner problems). *Under Assumption 3.1, Assumption 3.2 and Assumption 3.3, we have:*

$$D(N, \Delta N) \geq -\frac{1}{2}\|\big(\mathcal{P}(K_{out}K_{in}^\top) - \mathcal{P}(K_{out}K_{in}^\top E_N U)\big)(K_{out}r_N - \omega)\|_2^2, \tag{9}$$

*where $\mathcal{P}(X)$ is the orthogonal projection on $\text{range}(X)$, $E_N = \big((I - \eta BK_{in}^\top)^N - I\big)(BK_{in}^\top)^{-1}$ and $r_N = K_{in}^\top E_N(Bz_0 + c) + z_0$.*

---

The detailed proof is in Appendix A where we also cover bilevel optimization settings. We give closed-form expressions for $z_N(\theta) = K_{in}^\top E_N U\theta + r_N,$[2] $\ell(z_{N+\Delta N}(\theta^{\star,N}))$ and $D(N, \Delta N)$.

---

[2] Note that the overall model is therefore affine in the parameters.

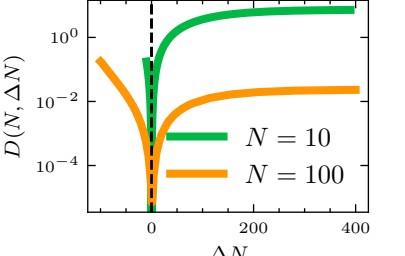 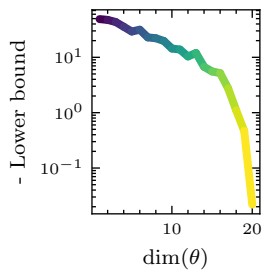 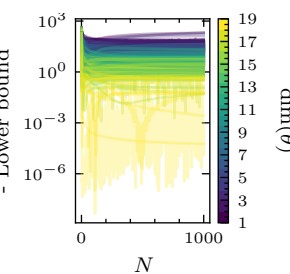

Figure 1: **Illustrations of the corollaries**. The left-hand side figure illustrates Corollary 1, i.e., the overparametrization regime. The black dashed line is at $\Delta N = 0$. We consider $z$ of dimension 5, $\theta$ of dimension 4, $F$ whose gradient is $f$ and $\ell$ convex but not strongly convex quadratic functions. The right-hand side figure illustrates Corollary 2, i.e. the average case. We report the Negative Lower bound, i.e. $\frac{1}{2}\|(\mathcal{P}(K_{\text{out}}K_{\text{in}}^\top) - \mathcal{P}(K_{\text{out}}K_{\text{in}}^\top E_N U))(K_{\text{out}}r_N - \omega)\|_2^2$, from Theorem 1. The inner and outer problems are strongly convex, and the dimension of $z$ is 20. The inner problem would be overparametrized in $\theta$ on average for dimension 20. We compute the lower bound for different inner optimization times and 20 seeds. In the left panel, we show the negative lower bound for $N = 100$ averaged over all the seeds for each dimension of $\theta$. In the right panel, we show the negative lower bounds for all seeds and for all $N$ for different dimensions of $\theta$.

This result states that it is possible to improve the loss by changing the number of iterations only up to a "point", and gives a quantitative result for this "point" (i.e. the lower bound).

The lower bound of $D(N, \Delta N)$ is independent of $\Delta N$ which means that for every $N$ the maximum decrease in loss achievable by changing the number of inner iterations at test time is bounded from below. This lower bound is the negative squared norm of a vector $(\mathcal{P}(K_{\text{out}}K_{\text{in}}^\top) - \mathcal{P}(K_{\text{out}}K_{\text{in}}^\top E_N U))(K_{\text{out}}r_N - \omega)$. Let us describe each component of this expression. $(K_{\text{out}}r_N - \omega)$ is a vector that encompasses elements from the inner problem and procedure $(r_N)$ and from the outer problem $(\omega, K_{\text{out}})$. $(\mathcal{P}(K_{\text{out}}K_{\text{in}}^\top) - \mathcal{P}(K_{\text{out}}K_{\text{in}}^\top E_N U))$ is a difference of projectors which measures how much $U$ is "not surjective relative to $K_{\text{out}}K_{\text{in}}^\top E_N$", and this is highlighted by the next two corollaries which explain in more details the role the surjectivity –i.e. the column rank– of $U$ in this lower bound. In appendix, Theorem I.11 explains how Theorem 1 can be transferred to non-affine functions $f$, provided that the initialization $z_0$ is close to the solution $z^*(\theta^*)$.

**Corollary 1** (Overparametrization in $\theta$). *Under Assumption 3.1, Assumption 3.2 and Assumption 3.3, if $U$ is surjective, we have for all $\Delta N$:*

$$\ell(z_{N+\Delta N}(\theta^{\star,N})) \geq \ell(z_N(\theta^{\star,N})), \tag{10}$$

*Proof.* When $U$ is surjective, $\mathcal{P}(K_{\text{out}}K_{\text{in}}^\top E_N U) = \mathcal{P}(K_{\text{out}}K_{\text{in}}^\top E_N)$. Further, $\forall N \geq N_0$, $E_N$ is invertible. Therefore, $\mathcal{P}(K_{\text{out}}K_{\text{in}}^\top E_N) = \mathcal{P}(K_{\text{out}}K_{\text{in}}^\top)$. We can conclude using Theorem 1. $\square$

In other words, when the inner model is sufficiently expressive, a regime called overparametrization, the inner variable is simply reparametrized with the outer variable, allowing the overall procedure to find the global minimum.

We numerically validate Corollary 1 with small-scale experiments on a quadratic bilevel optimization problem, and show the results in Figure 1 (left).

However, one can wonder what can be said when we are not in the overparametrized regime. The next corollary shows an example of what happens when the matrix $U$ is not overparametrized, and its entries are drawn i.i.d. from a Gaussian distribution. This last assumption is strong and this is why the next result cannot be considered as general but rather illustrative.

**Corollary 2** (Average case). *Under Assumption 3.1, Assumption 3.2 and Assumption 3.3, if $K_{in}$ is invertible and $\ell$ is strongly convex, we have:*

$$\mathbb{E}_{U \sim \mathcal{N}(0,I)}[D(N, \Delta N)] \geq -\frac{1}{2}(1 - \frac{\min(d_x, d_\theta)}{d_x})(\rho(K_{out})\|r_{max}\|_2^2 + \|\omega\|_2^2), \tag{11}$$

*with $\rho(K_{out})$ the spectral radius of $K_{out}$ and $\|r_{max}\|_2^2 \in \max_N \|r_N\|_2^2$.*

*Proof.* The proof relies on the computation of the expected value of the norm of the projection on the image of a matrix for a matrix with random coefficients. It is given in full in Appendix A □

An edge case of Corollary 2 is the situation where $d_x \leq d_\theta$, because the left term cancels. But this situation is on average equivalent to the overparametrization regime because we go from a $d_\theta$-dimensional space to a $d_x$-dimensional space. What Corollary 2 tells us is that as we get closer to overparametrization by increasing the dimension of the outer variable $\theta$, the expected loss decrease we could get by changing the number of inner iterations gets closer to 0.

We ran an experiment to illustrate Corollary 2 whose results are shown in Figure 1 (right). As expected, the lower bound of Theorem 1 does not vary significantly for different numbers of inner iterations. Moreover, we observe that the lower bound decreases in magnitude as the inner problem is more and more overparametrized in $\theta$. In cases where the inner or outer problems are not strongly convex, the lower bound can be 0 even before $U$ is surjective. We show such cases in Appendix H.

In order to understand in which regime fall DEQs and meta-learning we need to understand to which extent the inner problem is close to overparametrization in the outer variable $\theta$. If one wants to be close to overparametrization on average, this requires an outer variable $\theta$ of dimension $d_z \times n$, where $n$ is the number of samples. While this number is usually prohibitively large, it is a common assumption in deep learning setups to assume that one can overfit the training data (Li & Liang, 2018; Du et al., 2018; Arora et al., 2019). However, in the case of meta-learning, since we are learning on multiple tasks at the same time, it is impossible a priori to overfit all the tasks at the same time since they might have contradicting objectives. Therefore, we expect DEQs to have a hard time benefiting from more iterations, while there is room for meta-learning to do so.

We stress that these results do not cover the generalization to a test dataset $\mathcal{D}_{\text{test}}$. Indeed, the quantity $D(N, \Delta N)$ only monitors the change in training loss achieved by changing the number of inner iterations. However, $D(N, \Delta N)$ is a good proxy for the change in test loss provided a large enough training data set. We provide a generalization result in Corollary D.1 (in Appendix D) for DEQs stacked sample-wise with some additional smoothness assumptions.

**Implicit differentiation for affine inner problems** It can be noted that Theorem 1 considers $\theta^{\star,N}$ as one of the solutions to (P). However, most of the time in practice (Bai et al., 2019; Rajeswaran et al., 2019), the optimization is actually performed using the approximate implicit differentiation gradients, rather than the true unrolled gradients in order to have a memory-efficient training. For an inner function $f$, this descent, with constant step size $\alpha_N$ (i.e. independent of $T$), would typically be written as the following:

$$
\begin{aligned}
\theta_{\text{IFT}}^{T+1,N} &= \theta_{\text{IFT}}^{T,N} - \alpha_N p_N(\theta_{\text{IFT}}^{T,N}) \\
\text{where } p_N(\theta) &= -(\partial_z f(z_N, \theta)^\dagger \partial_\theta f(z_N, \theta))^\top \nabla\ell(z_N),
\end{aligned}
\tag{12}
$$

where † stands for the pseudo-inverse. This equation is a practical implementation of (3) in the case where the Jacobian is not necessarily invertible, and we use an approximate inner solution $z_N$. This formula is heuristic and does not necessarily provide a descent direction for arbitrary $N$. We defer the proofs of the following results to Appendix C.

**Lemma 1** (Convergence of the practical IFT gradient descent). *Under Assumption 3.1, Assumption 3.2 and Assumption 3.3, with $\ell$ strongly convex, $\exists N_0$ such that $\forall N > N_0$, $\exists \alpha_N > 0$ such that the sequences $\theta_{\text{IFT}}^{T,N}$ converge to a value denoted $\theta_{\text{IFT}}^{\star,N}$, dependent only on the initialization.*

Therefore, practical optimal solutions, denoted $\theta_{\text{IFT}}^{\star,N}$, are solutions to the following root problem:

$$
(\partial_z f(z_N(\theta), \theta)^\dagger \partial_\theta f(z_N(\theta), \theta))^\top \nabla\ell(z_N(\theta)) = 0
\tag{13}
$$

**Theorem 2** (Equivalence of IFT and unrolled solutions for affine inner problems). *Under Assumption 3.1, Assumption 3.2 and Assumption 3.3, with $\ell$ strongly convex and $U$ surjective, we have:*

$$
\theta_{IFT}^{\star,N} = \theta^{\star,N}
\tag{14}
$$

For overparametrized cases, where $U$ is surjective, this means that Corollary 1 is valid for IFT-based implicit deep learning. Therefore, this shows that Implicit Deep Learning trained with IFT is not less prone to I2O than if it is trained through unrolling in this case. We confirm this empirically in Section 5 for practical cases with DEQs.

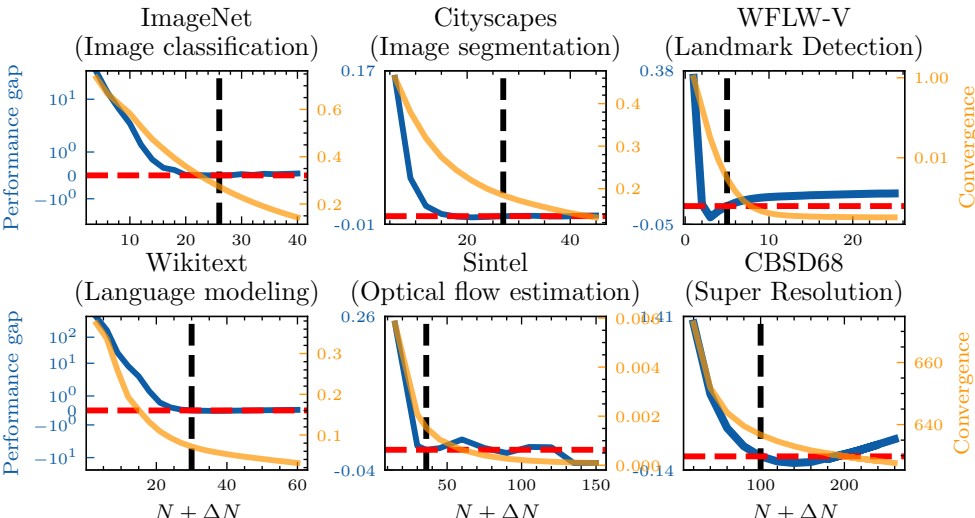

Figure 2: **I2O for DEQs**: Test performance gap (lower is better) using $N + \Delta N$ inner iterations at inference compared to using $N$ inner iterations. This performance gap reaches 0 after reaching $N$ the number of iterations used during training. The black dashed line is at $\Delta N = 0$, i.e. the training number of inner iterations. The red dashed line is at 0. For ImageNet, the performance is measured using the top-1 error rate (%), for Cityscapes it is measured using the negative mean IoU, for WikiText it is measured using the perplexity and for optical flow it is measured using the average EPE. Note that after $N$ iterations, getting closer to convergence (plotted as $\|z_{N+\Delta N}(\theta^{\star,N}) - f(z_{N+\Delta N}(\theta^{\star,N}), \theta^{\star,N})\|$) does not bring a performance benefit.

## 4 THE EMPIRICAL PHENOMENON OF INNER ITERATIONS OVERFITTING

In order to validate the results from our theoretical analysis in realistic cases, we explore the I2O phenomenon for DEQs and (i)MAML experiments, two settings that highlight the different regimes from our theoretical results.

**DEQs** We conduct experiments on pre-trained DEQs in various successful applications. The evaluation is unchanged except for the number of inner iterations, which is the same for training and inference in the typical deq[3] library (Bai et al., 2019). The settings we cover are text completion on Wikitext (Bai et al., 2019; Merity et al., 2017), large-scale image classification on ImageNet (Bai et al., 2020; Deng et al., 2009), image segmentation on Cityscapes (Bai et al., 2020; Cordts et al., 2016), optical flow estimation on Sintel (Bai et al., 2022a; Butler et al., 2012), single image super resolution (×3) on CBSD68 (Zou et al., 2023; Martin et al., 2001), and landmark detection on WFLW-V (Micaelli et al., 2023). We report the test performance gap in Figure 2, i.e. the difference between the test performance for $N + \Delta N$ inner iterations and the test performance for $N$ inner iterations and give more details on the experiments in Appendix G. The test performance is always cast as lower is better.

The figure shows that for all four cases, the performance of the model does not improve when increasing the number of iterations at test time (blue lines) and therefore improving convergence (orange lines). Once $\Delta N$ becomes positive, the performance plateaus and in the worst cases such as landmark detection or super resolution, it deteriorates. This highlights I2O for DEQs, which are overparametrized (e.g. reaching 90% training accuracy on ImageNet compared to 80% test accuracy).

Similar observations can be made on the training performance and on training and test losses, as demonstrated in Figure H.2 and Figure H.1 in the appendix. This shows that while our theoretical results uncover this phenomenon in simple cases for the training loss only, it is observable for more complex setups, even for the test performance gap, further validating our result in Theorem I.10.

**(i)MAML** Unlike DEQs, (i)MAML is less prone to I2O. For example, Chen et al. (2020) meta-train a network with $N = 100$ inner steps, and meta-test it with $N + \Delta N = 10000$ inner steps. We run

---
[3]github.com/locuslab/deq

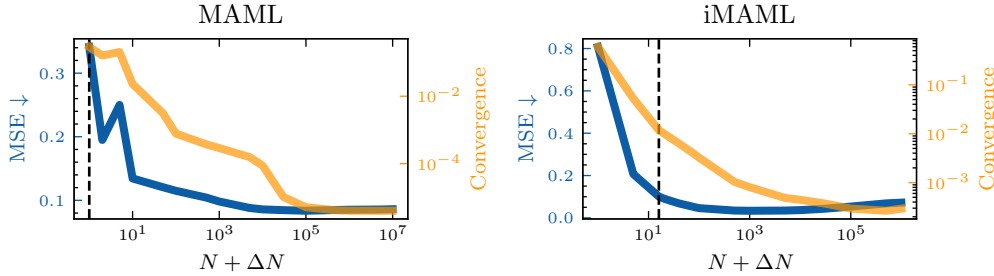

Figure 3: **(i)MAML is less prone to I2O**: Test MSE using $N + \Delta N$ inner iterations at inference. The black dashed line is at $\Delta N = 0$, i.e. the training number of inner iterations. Note the log-scale for the x-axis. Note that the best MSE is reached before convergence.

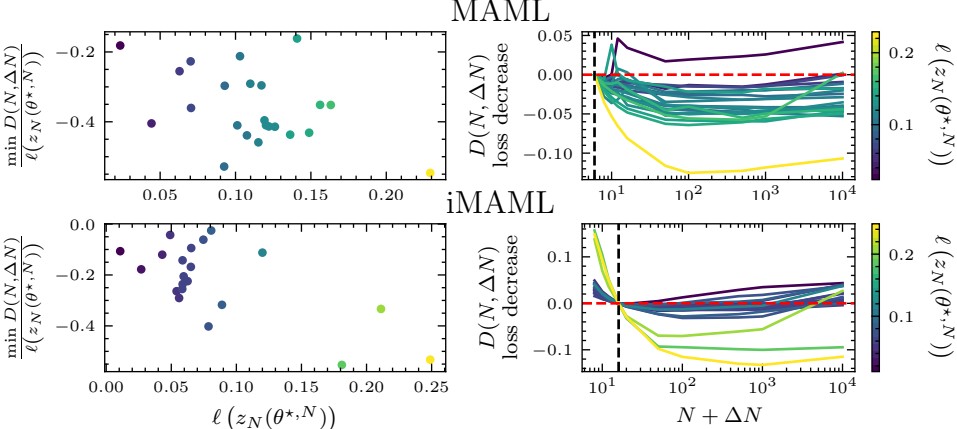

Figure 4: **The impact of overparametrization for (i)MAML**: Normalized training loss increase $D(N, \Delta N)/\ell(z_N(\theta^{\star,N}))$ for different levels of overparametrization. This level of overparametrization is measured by the loss on the training set for $N$ inner iterations, $\ell(z_N(\theta^{\star,N}))$. Generally, the more meta-batches there are in the training set, the higher this loss. The black dashed line is at $\Delta N = 0$, i.e. the training number of inner iterations. The red dashed line is at $D = 0$, i.e. no increase. Note that the more overparametrized the model, the higher $D(N, \Delta N)/\ell(z_N(\theta^{\star,N}))$ is. The Pearson correlation coefficient for the top left plot is $-0.4$ with a p-value of $0.03$, while for the bottom left plot it is $-0.6$ with a p-value of $0.007$.

experiments on the synthetic sinusoids regression task introduced by Finn et al. (2017) and confirm that I2O is much less prevalent in (i)MAML. Figure 3 shows the variation in test performance for the meta-learning task when changing the number of inner iterations. More details on this experiment are given in Appendix G.

We see that while the best performance at test time is achieved for a number of inner iterations much larger than the one used during training, this improvement is bounded from below as predicted by Theorem 1: the best MSE is reached before convergence. As for DEQs, reaching convergence does not provide the best performance.

Further, we highlight the effect of overparametrization in the iMAML case. As the overparametrization is not easy to measure, we use overfitting as a proxy: the better the model is at overfitting, the more overparametrized it is. Figure 4 shows the normalized training loss increase for various training set sizes for meta-learning, from 1 meta-batch –easy to overfit– to 25 meta-batches –harder to overfit. We observe that the better the model is at overfitting the training dataset –i.e. low $\ell(z_N(\theta^{\star,N}))$– the less it benefits from more iterations at inference –i.e. higher $D(N, \Delta N)$, even if normalized. We conduct this experiment on the training loss, as the test data is too different from the training one with so few meta-batches.

**Upwards generalization and path-independence** Our results might seem to be in contradiction with the empirical findings reported by Anil et al. (2022). They study how the test-time performance

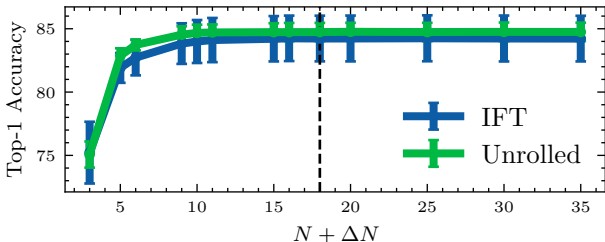

Figure 5: **Stability of unrolled/IFT trained networks**: We compare how stable networks trained with either unrolling or IFT are to the choice of the number of inner iterations at inference. The task is image classification on CIFAR-10. The networks were trained with 18 iterations. Note that the IFT-trained network does not appear more stable than its unrolled counterpart.

is affected by an increase in the number of inner iterations at inference, but unlike us consider harder problems. They correlate the capacity of DEQs to benefit from more test-time inner iterations with a property termed path-independence. A DEQ is said to be path-independent if for a given couple $\theta, x_i$, there exists only one root of $f(z, \theta, x_i)$.

We highlight that our theoretical analysis is also valid for path-independent DEQs, which also suffer from I2O in-distribution. Typically, when $K_{\text{in}}^\top = I$ and we have a fully invertible affine DEQ layer, there is only a single root $z^\star(\theta) = -B^{-1}(U\theta + c)$ (see Assumption 3.2) and this is covered by Theorem 1. However, in our experiments, we consider in-distribution generalization, rather than upwards generalization, i.e., an out-of-distribution setting where an explicit difficulty parameter is set higher at test-time than during training, which explains why our results and those of Anil et al. (2022) are not in contradiction.

## 5   ROBUSTNESS OF THE NETWORKS OBTAINED WITH IFT GRADIENT DESCENT TO I2O

Theorem 2 shows that the way the hypergradient is computed for implicit deep learning, IFT or unrolling, does not impact whether it suffers from I2O in our simplified setting. Still, one might wonder whether the effect of I2O is stronger for one or the other in practical cases. Gilton et al. (2021) suggest that I2O is much more prevalent for unrolling, highlighting a huge drop in performance when more iterations are used during inference. However, in their experiment, the network trained with unrolling has an effective depth much smaller than its IFT-trained counterpart –10 fixed-point iterations vs 50 Anderson acceleration iterations. Indeed, because IFT gradient descent enables memory-free training, it is possible to train networks with a very large effective depth, while it is harder for unrolled networks.

We tried to test whether this conclusion still holds when training networks with unrolling and IFT with the same depth. In our experiment, both networks use Broyden's method to solve the inner problem. Therefore, for the unrolled network, we backpropagate through the Broyden iterates which has a high memory requirement. For this reason, we can only train relatively small networks for image classification on CIFAR-10 (Krizhevsky, 2009). In Figure 5 we compare the stability of the two networks for different numbers of inner iterations averaging performance over 10 seeds. We observe that there is no gain of stability when using IFT gradient descent over unrolling.

## 6   CONCLUSION

In this work, we challenge one common assumption about DEQs: the possibility to select their computational budget at inference. We showed that not only do DEQs exhibit a phenomenon we termed inner iterations overfitting (I2O), that we proved to be grounded in theory for simple models, they also do not appear to have more stability than unrolled networks. We highlight that this does not mean that DEQs should not be used: the $O(1)$ memory requirement during training is a strong point that enables the training of very deep networks not trainable with unrolling.

Since we noted that eventually, one wants to use DEQs with a fixed number of iterations, a question that remains is: can we bias the hypergradient so that it does take into account the number of iterations made in a more direct way, somehow making it more similar to the unrolled gradient?

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

## A   PROOF OF MAIN RESULT AND COROLLARIES

In order to prove the main result, we will rely on the notion of time-invertible linear procedures.

**Definition A.1.** (Time-invertible linear procedure) A time-invertible linear procedure is a sequence $z_N(\theta)$ such that there exists $N_0 < \infty$, $K_{\text{in}} \in \mathbb{R}^{d_x \times d_z}$, $U \in \mathbb{R}^{d_x \times d_\theta}$, $E_N \in \mathbb{R}^{d_x \times d_x}$, $r_N \in \mathbb{R}^{d_z}$ such that it can be written as:

$$z_N(\theta) = K_{\text{in}}^\top E_N U\theta + r_N, \tag{15}$$

and $\forall N \geq N_0$, where $E_N$ is invertible.

Let us now move on to the main components of the proof. We need first to get an expression for the iterates of the fixed-point iterations for an affine function.

**Lemma A.2.** *Let us assume $f(z) = K^\top(Bz + c)$, with $K, B \in \mathbb{R}^{d_x \times d_x}$ and $c \in \mathbb{R}^{dx}$. We further assume that $BK^\top$ has eigenvalues with positive real part and that $K$ is surjective. Then the iterates of the fixed-point iteration method with fixed step size $\eta$ have the following expression:*

$$z_N = K^\top \left( \left( I - \eta BK^\top \right)^N - I \right) (BK^\top)^{-1}(Bz_0 + c) + z_0, \tag{16}$$

*with $z_0$ the initialization of the procedure.*

*Proof.* We have:

$$z_{N+1} = z_N - \eta K^\top(Bz_N + c) \tag{17}$$
$$= (I - \eta K^\top B)z_N - \eta K^\top c \tag{18}$$

First let us check that $z_N - z_0$ is in the range of $K^\top$ for all $N$. We proceed by recurrence starting with $N = 0$, which is obviously true because $z_0 - z_0 = 0$ is the range of $K^\top$. If $z_N - z_0$ is in the range of $K^\top$, there exists $x$ such that $z_N - z_0 = K^\top x$. Then we have

$$z_{N+1} = z_N - \eta K^\top B z_N - \eta K^\top c \tag{19}$$
$$z_{N+1} - z_0 = z_N - z_0 - \eta K^\top B z_N - \eta K^\top c \tag{20}$$
$$z_{N+1} - z_0 = K^\top x - \eta K^\top B z_N - \eta K^\top c \tag{21}$$
$$z_{N+1} - z_0 = K^\top (x - \eta B z_N - \eta c) \tag{22}$$

Therefore $z_{N+1} - z_0$ is also in the range of $K^\top$ and we conclude that $z_N - z_0$ is in the range of $K^\top$ for all $N$. We then introduce $y_N$ such that $z_N - z_0 = K^\top y_N$. The following recurrence then holds:

$$z_{N+1} - z_0 = z_N - z_0 - \eta K^\top B z_N - \eta K^\top c \tag{23}$$
$$K^\top y_{N+1} = K^\top y_N - \eta K^\top B z_N - \eta K^\top c \tag{24}$$
$$K^\top y_{N+1} = K^\top (y_N - \eta B z_N - \eta c) \tag{25}$$
$$y_{N+1} = y_N - \eta B z_N - \eta c \tag{26}$$
$$y_{N+1} = y_N - \eta B(z_N - z_0) - \eta B z_0 - \eta c \tag{27}$$
$$y_{N+1} = y_N - \eta B K^\top y_N - \eta B z_0 - \eta c \tag{28}$$
$$y_{N+1} = \left( I - \eta B K^\top \right) y_N - \eta B z_0 - \eta c \tag{29}$$

The expression of $y_N$ is then for $y_0 = 0$:

$$y_N = \left( I - \eta BK^\top \right)^N (BK^\top)^{-1}(Bz_0 + c) - (BK^\top)^{-1}(Bz_0 + c) \tag{30}$$
$$= \left( \left( I - \eta BK^\top \right)^N - I \right) (BK^\top)^{-1}(Bz_0 + c) \tag{31}$$

Therefore the expression of $z_N$ is:

$$z_N - z_0 = K^\top \left( \left( I - \eta BK^\top \right)^N - I \right) (BK^\top)^{-1}(Bz_0 + c) \tag{32}$$
$$z_N = K^\top \left( \left( I - \eta BK^\top \right)^N - I \right) (BK^\top)^{-1}(Bz_0 + c) + z_0 \tag{33}$$

$\square$

**Lemma A.3.** *Under Assumption 3.1 and Assumption 3.2, the inner procedure (i.e. the fixed-point iteration method) is a time-invertible linear procedure with $E_N = \left((I - \eta BK_{in}^\top)^N - I\right)(BK_{in}^\top)^{-1}$ and $r_N = K_{in}^\top E_N(Bz_0 + c) + z_0$, with $z_0$ the initialization of the procedure.*

*Proof.* Using Lemma A.2, we have the expression of $z_N(\theta)$:

$$z_N(\theta) = K_{in}^\top \left((I - \eta BK_{in}^\top)^N - I\right)(BK_{in}^\top)^{-1}(U\theta + c + Bz_0) + z_0 \tag{34}$$

$$= K_{in}^\top \left((I - \eta BK_{in}^\top)^N - I\right)(BK_{in}^\top)^{-1}U\theta \tag{35}$$

$$+ K_{in}^\top \left((I - \eta BK_{in}^\top)^N - I\right)(BK_{in}^\top)^{-1}(c + Bz_0) + z_0 \tag{36}$$

$$= K_{in}^\top E_N U\theta + r_N, \tag{37}$$

with $E_N = \left((I - \eta BK_{in}^\top)^N - I\right)(BK_{in}^\top)^{-1}$ and $r_N = K_{in}^\top E_N(c + Bz_0) + z_0$. We now need to prove that $E_N$ is invertible for $N$ sufficiently large. Since the fixed-point iterations converge, $(I - \eta BK_{in}^\top)^N$ goes to 0, and therefore $\left((I - \eta BK_{in}^\top)^N - I\right)$ goes to $-I$ which means that for $N$ sufficiently large the latter is invertible. We conclude by noticing that $E_N$ is invertible as the product of two invertible matrices. $\qquad\square$

We restate the main theorem here for convenience, before proving it.

**Theorem 1** (Inner Iterations Overfitting for affine inner problems). *Under Assumption 3.1, Assumption 3.2 and Assumption 3.3, we have:*

$$D(N, \Delta N) \geq -\frac{1}{2}\|\left(\mathcal{P}(K_{out}K_{in}^\top) - \mathcal{P}(K_{out}K_{in}^\top E_N U)\right)(K_{out}r_N - \omega)\|_2^2, \tag{9}$$

*where $\mathcal{P}(X)$ is the orthogonal projection on $\mathrm{range}(X)$, $E_N = \left((I - \eta BK_{in}^\top)^N - I\right)(BK_{in}^\top)^{-1}$ and $r_N = K_{in}^\top E_N(Bz_0 + c) + z_0$.*

*Proof.* We prove here the result directly for a time-invertible linear procedure thanks to Lemma A.3:

$$z_N(\theta) = K_{in}^\top E_N U\theta + r_N = A_N\theta + r_N, \text{ with } A_N = K_{in}^\top E_N U \tag{38}$$

We can then write:

$$\ell(z_N(\theta)) = \frac{1}{2}\|K_{out}z_N(\theta) - \omega\|_2^2 \tag{39}$$

$$= \frac{1}{2}\|K_{out}A_N\theta + K_{out}r_N - \omega\|_2^2 \tag{40}$$

$$= \frac{1}{2}\|K_{out}A_N\theta + \mathcal{P}(K_{out}A_N)(K_{out}r_N - \omega)\|_2^2 + \frac{1}{2}\|\mathcal{P}((K_{out}A_N)^\perp)(K_{out}r_N - \omega)\|_2^2 \tag{41}$$

$$(K_{out}, A_N, r_N, \omega) \mapsto \lim_{t\to\infty}\theta_{t+1} = \theta_t - (K_{out}A_N)^\top(K_{out}A_N\theta_t + K_{out}r_N - \omega) \tag{42}$$

where the last equation holds because the two terms are orthogonal. And we have:

$$K_{out}K_{in}^\top E_N U\theta^{\star,N} = -\mathcal{P}(K_{out}A_N)(K_{out}r_N - \omega) \tag{43}$$

$$E_N U\theta^{\star,N} = -(K_{out}K_{in}^\top)^\dagger \mathcal{P}(K_{out}A_N)(K_{out}r_N - \omega) + \theta_{\ker(K_{out}K_{in}^\top)} \tag{44}$$

$$U\theta^{\star,N} = -E_N^{-1}(K_{out}K_{in}^\top)^\dagger \mathcal{P}(K_{out}A_N)(K_{out}r_N - \omega) + E_N^{-1}\theta_{\ker(K_{out}K_{in}^\top)} \tag{45}$$

Plugging that into the outer loss for a different inner iterations number $N' = N + \Delta N$ and using $C = K_{out}K_{in}^\top$, $M(N', N) = E_{N'}E_N^{-1}$ and $v_N = (K_{out}r_N - \omega)$:

$$\ell(z_{N'}(\theta^{\star,N})) = \frac{1}{2}\|CE_{N'}U\theta^{\star,N} + K_{out}r_{N'} - \omega\|_2^2 \tag{46}$$

$$= \frac{1}{2}\| - CM(N', N)C^\dagger\mathcal{P}(K_{out}A_N)v_N + v_{N'} + CM(N', N)\theta_{\ker(C)}\|_2^2 \tag{47}$$

$$= \frac{1}{2}\| - CM(N', N)C^\dagger\mathcal{P}(K_{out}A_N)v_N + \mathcal{P}(C)v_{N'} + CM(N', N)\theta_{\ker(C)}\|_2^2 \tag{48}$$

$$+ \frac{1}{2}\|\mathcal{P}(C^\perp)v_{N'}\|_2^2$$

For $N' = N$, i.e. $\Delta N = 0$, since $M(N', N) = I$, we have:

$$\ell(z_N(\theta^{\star,N})) = \frac{1}{2}\| - CC^\dagger \mathcal{P}(K_{\text{out}} A_N)v_N + \mathcal{P}(C)v_N + C\theta_{\ker(C)}\|_2^2 + \frac{1}{2}\|\mathcal{P}(C^\perp)v_N\|_2^2 \quad (49)$$

$$= \frac{1}{2}\| - CC^\dagger \mathcal{P}(K_{\text{out}} A_N)v_N + \mathcal{P}(C)v_N\|_2^2 + \frac{1}{2}\|\mathcal{P}(C^\perp)v_N\|_2^2 \quad (50)$$

$$= \frac{1}{2}\| - \mathcal{P}(C)\mathcal{P}(CE_N U)v_N + \mathcal{P}(C)v_N\|_2^2 + \frac{1}{2}\|\mathcal{P}(C^\perp)v_N\|_2^2 \quad (51)$$

$$= \frac{1}{2}\| - \mathcal{P}(CE_N U)v_N + \mathcal{P}(C)v_N\|_2^2 + \frac{1}{2}\|\mathcal{P}(C^\perp)v_N\|_2^2 \quad (52)$$

$$= \frac{1}{2}\| \left(\mathcal{P}(C) - \mathcal{P}(CE_N U)\right) v_N\|_2^2 + \frac{1}{2}\|\mathcal{P}(C^\perp)v_N\|_2^2 \quad (53)$$

Finally, subtracting the two expressions, we have:

$$D(N, \Delta N) = \ell(z_{N+\Delta N}(\theta^{\star,N})) - \ell(z_N(\theta^{\star,N})) \quad (54)$$

$$\geq -\frac{1}{2}\| \left(\mathcal{P}(C) - \mathcal{P}(CE_N U)\right) v_N\|_2^2 \quad (55)$$

$$= -\frac{1}{2}\| \left(\mathcal{P}(K_{\text{out}} K_{\text{in}}^\top) - \mathcal{P}(K_{\text{out}} K_{\text{in}}^\top E_N U)\right) (K_{\text{out}} r_N - \omega)\|_2^2 \quad (56)$$

because $\mathcal{P}(C^\perp)v_N = \mathcal{P}(C^\perp)v_{N'} = -\mathcal{P}(C^\perp)\omega$. $\qquad \square$

**Lemma A.4.** *For $v \in \mathbb{R}^p$, and $\mathcal{P}(X)$ the orthogonal projection onto $\mathrm{range}(X)$ for $X \in \mathbb{R}^{p \times d}$, we have*

$$\mathbb{E}_{X|X_{ij} \sim \mathcal{N}(0,1)}\|\mathcal{P}(X)v\|_2^2 = \frac{d}{p}\|v\|_2^2. \quad (57)$$

*Proof.* The proof is taken from zhm1995 (2019). Let $X = UDV^\top$ a SVD of $X$, with $U \in \mathbb{R}^{d \times d}$ orthogonal, $D$ diagonal with positive entries, and $V \in \mathbb{R}^{p \times d}$ orthogonal, i.e. such that $V^\top V = I_d$. Since the entries of $X$ are drawn i.i.d. from a normal distribution, the singular values of $X$ are non-zeros with probability 1, and we have $\mathcal{P}(X) = X^\top (XX^\top)^{-1}X = VV^\top$. Since the distribution of $V$ is right invariant, the distribution of $\|\mathcal{P}(X)v\|_2^2$ depends only on $\|v\|_2^2$.[4] Thus, $\mathbb{E}_{X|X_{ij} \sim \mathcal{N}(0,1)}\|\mathcal{P}(X)v\|_2^2 = \|v\|_2^2 \mathbb{E}_{X|X_{ij} \sim \mathcal{N}(0,1)}\mathcal{P}(X)_{11}$, taking $v = (\|v\|, 0, \ldots, 0)$ (i.e. we chose a basis where the first vector is 0, and if $v$ is 0 the problem is trivial). By symmetry, we have $\mathbb{E}_{X|X_{ij} \sim \mathcal{N}(0,1)}\mathcal{P}(X)_{11} = \mathbb{E}_{X|X_{ij} \sim \mathcal{N}(0,1)} \mathrm{tr}(\mathcal{P}(X))/p$. And we have $\mathrm{tr}(\mathcal{P}(X)) = \mathrm{tr}(VV^\top) = \mathrm{tr}(V^\top V) = d$. $\qquad \square$

**Corollary 2** (Average case). *Under Assumption 3.1, Assumption 3.2 and Assumption 3.3, if $K_{in}$ is invertible and $\ell$ is strongly convex, we have:*

$$\mathbb{E}_{U \sim \mathcal{N}(0,I)} [D(N, \Delta N)] \geq -\frac{1}{2}(1 - \frac{\min(d_x, d_\theta)}{d_x})(\rho(K_{out})\|r_{max}\|_2^2 + \|\omega\|_2^2), \quad (11)$$

*with $\rho(K_{out})$ the spectral radius of $K_{out}$ and $\|r_{max}\|_2^2 \in \max_N \|r_N\|_2^2$.*

*Proof.* From Theorem 1, we have for $N \geq N_0$:

$$D(N, \Delta N) \geq -\frac{1}{2}\| \left(\mathcal{P}(K_{\text{out}} K_{\text{in}}^\top) - \mathcal{P}(K_{\text{out}} K_{\text{in}}^\top E_N U)\right) (K_{\text{out}} r_N - \omega)\|_2^2 \quad (58)$$

When the outer problem is strongly convex, we have $K_{\text{out}}$ is an invertible matrix. Therefore, $\mathcal{P}(K_{\text{out}} K_{\text{in}}^\top) = I$, and $K_{\text{out}} K_{\text{in}}^\top E_N = B_N$ is invertible for $N \geq N_0$. Using the notation

---

[4]Right invariance means, for each fixed $p \times p$ orthogonal matrix $B$, the matrix $VB$ is distributed the same way $V$ is. It is a consequence of the rotational symmetry of the original normal distribution.

$v_N = (K_\text{out} r_N - \omega)$, if $d_x < d_\theta$, we have:

$$\mathbb{E}_{U \sim \mathcal{N}(0,I)} D(N, \Delta N) \geq -\frac{1}{2} \mathbb{E}_{U \sim \mathcal{N}(0,I)} \| \left( \mathcal{P}(K_\text{out} K_\text{in}^\top) - \mathcal{P}(K_\text{out} K_\text{in}^\top E_N U) \right) (K_\text{out} r_N - \omega) \|_2^2 \tag{59}$$

$$\geq -\frac{1}{2} \mathbb{E}_{U \sim \mathcal{N}(0,I)} \| \left( I - \mathcal{P}(B_N U) \right) v_N \|_2^2 \tag{60}$$

$$\geq -\frac{1}{2} \mathbb{E}_{U \sim \mathcal{N}(0,I)} v_N^\top \left( I - \mathcal{P}(B_N U) \right)^\top \left( I - \mathcal{P}(B_N U) \right) v_N \tag{61}$$

$$= -\frac{1}{2} \mathbb{E}_{U \sim \mathcal{N}(0,I)} v_N^\top \left( I - \mathcal{P}(B_N U) \right) \left( I - \mathcal{P}(B_N U) \right) v_N \tag{62}$$

$$= -\frac{1}{2} \mathbb{E}_{U \sim \mathcal{N}(0,I)} v_N^\top \left( I - \mathcal{P}(B_N U) \right) v_N \tag{63}$$

$$= -\frac{1}{2} \mathbb{E}_{U \sim \mathcal{N}(0,I)} v_N^\top v_N - v_N^\top \mathcal{P}(B_N U) v_N \tag{64}$$

$$= -\frac{1}{2} \mathbb{E}_{U \sim \mathcal{N}(0,I)} \|v_N\|_2^2 - v_N^\top \mathcal{P}(B_N U) v_N \tag{65}$$

$$= -\frac{1}{2} \mathbb{E}_{U \sim \mathcal{N}(0,I)} \|v_N\|_2^2 - v_N^\top \mathcal{P}(B_N U) \mathcal{P}(B_N U) v_N \tag{66}$$

$$= -\frac{1}{2} \mathbb{E}_{U \sim \mathcal{N}(0,I)} \|v_N\|_2^2 - v_N^\top \mathcal{P}(B_N U)^\top \mathcal{P}(B_N U) v_N \tag{67}$$

$$= -\frac{1}{2} \mathbb{E}_{U \sim \mathcal{N}(0,I)} \|v_N\|_2^2 - \|\mathcal{P}(B_N U) v_N\|_2^2 \tag{68}$$

$$= -\frac{1}{2} \mathbb{E}_{U' \sim \mathcal{N}(0,I)} \|v_N\|_2^2 - \|\mathcal{P}(U') v_N\|_2^2 \tag{69}$$

$$= -\frac{1}{2} \|v_N\|_2^2 - \mathbb{E}_{U' \sim \mathcal{N}(0,I)} \|\mathcal{P}(U') v_N\|_2^2 \tag{70}$$

$$= -\frac{1}{2} \|v_N\|_2^2 - \frac{d_\theta}{d_x} \|v_N\|_2^2 \tag{71}$$

$$= -\frac{1}{2} \left( 1 - \frac{d_\theta}{d_x} \right) \|v_N\|_2^2 \tag{72}$$

We can conclude using the triangular inequality and the definition of the spectral radius on $\|v_N\|_2^2$, and using Corollary 1 to go from $\frac{d_\theta}{d_x}$ to $\frac{\min(d_x, d_\theta)}{d_x}$. $\square$

## B HOW RELEVANT IS THE AFFINE INNER PROBLEM PARAMETERIZATION?

In order to derive our analysis, we assumed in Assumption 3.2 that the expression of the inner problem root-defining function followed a certain factorization $f(z, \theta) = K_\text{in}^\top (Bz + U\theta + c)$. We show that this form is satisfied by two classes of problems: affine DEQs and meta-learning a linear model.

We first tackle the case of affine DEQs. To do so, let us first establish the following lemma:

**Lemma B.1.** *If $f$ is an affine function of the form $f(z) = Az + c$, and the fixed-point iteration method with fixed step size $\eta$ to find its root converges for any initialization $z_0$, then $f$ can be factorized, i.e. $\exists K, \Gamma, \gamma$ such that $f(z) = K^\top \Gamma z + K^\top \gamma$ and $K$ is surjective. Moreover, $\Gamma K^\top$ has eigenvalues with positive real part.*

*Proof.* If the fixed-point iteration method converges, then it means that $f$ has a root denoted $z^\star$. This root verifies $Az^\star = -c$, so it means that $c$ is in the range of $A$. Furthermore, we can write the low-rank factorization of $A$ as $K^\top \Gamma$ with $K$ surjective. Because $c$ is in the range of $A$ it is also in the range of $K^\top$. We can denote $c = K^\top \gamma$. Using Lemma A.2, we have the expression of $z_N$:

$$z_N = K^\top \left( \left( I - \eta \Gamma K^\top \right)^N - I \right) (\Gamma K^\top)^{-1} (\Gamma z_0 + \gamma) + z_0 \tag{73}$$

Since $z_N$ converges for any $z_0$, this means that the largest eigenvalue of $(I - \eta \Gamma K^\top)$ is bounded by 1 in magnitude. We can choose $\eta$ as $\frac{1}{\lambda_\text{max} + \epsilon}$ to realize this if all the eigenvalues of $\Gamma K^\top$ have a positive real part. $\square$

Similarly we have the following result:

**Proposition 1.** *Let us assume that $f$ is affine in $z$ and $\theta$. If the fixed-point iteration method with fixed step size $\eta$ converges for $f$, then it satisfies Assumption 3.2.*

We now move on to meta-learning a linear model with quadric loss which we show to be a special case of the above.

**Proposition 2.** *. If the task specific regularized loss for iMAML is a convex qudratic function and can be written as:*

$$F(z, \theta) = \frac{1}{2}\|Xz - y\|_2^2 + \lambda\|z - \theta\|_2^2, \tag{74}$$

*with $\mathcal{X}_{train} = (X, y)$ the task training set, and $\lambda$ the meta regularization parameter, and gradient descent with fixed step size $\eta$ converges to minimize $F$ in $z$, then $\nabla_z F$ satisfies Assumption 3.2.*

*Proof.* Since $F$ is a quadratic function of $z$ and $\theta$, $\nabla F$ is an affine function and the gradient descent on $F$ with fixed step size $\eta$ corresponds to the fixed-point iteration method for $\nabla F$. We can conclude using Proposition 1 $\qquad\square$

**Stacking DEQ sample-wise problems**   Our analysis is conducted for the problem obtained when stacking all the independent sample-wise problems. Indeed, for DEQs, for each sample $x_i$ we have $f(z, \theta, x_i) = K_{\text{in}}(x_i)^\top(B(x_i)z + U(x_i)\theta + c(x_i))$. However, Assumption 3.2 applies to the problem from the dataset point of view. This can be related when stacking together the individual sample-wise components as follows:

$$\mathbf{z}_N(\theta) = [z_N(\theta, x_1), \ldots, z_N(\theta, x_n)]^\top \tag{75}$$

$$\mathbf{U} = [U(x_1), \ldots, U(x_n)] \tag{76}$$

$$\mathbf{B} = \begin{bmatrix} B(x_1) & & \\ & \ddots & \\ & & B(x_n) \end{bmatrix} \tag{77}$$

$$\mathbf{K}_{\text{in}} = \begin{bmatrix} K_{\text{in}}(x_1) & & \\ & \ddots & \\ & & K_{\text{in}}(x_n) \end{bmatrix} \tag{78}$$

$$\mathbf{c} = [c(x_1), \ldots, c(x_n)]. \tag{79}$$

A similar remark applies to Assumption 3.3.

**Difference with the parametrization of Winston & Kolter (2020)**   As stated in Section 3, the parametrization we chose is different from that of Winston & Kolter (2020). Indeed, they chose to use:

$$f(z, W, U, b, x_i) = \sigma(Wz + Ux + b), \tag{80}$$

where the parameters $(W, U, b)$ are the equivalent of $\theta$. In plain text, the difference with our parameterization, other than the pointwise nonlinearity $\sigma$, is that they allow multiplicative interactions between the output $z$ and part of the parameters $W$. This type of parametrization is well suited to anlyzing convergence, however in our case, it would lead to a dyanmical system with an overly complex trajectory not fit for our analysis. To illustrate this point, let us assume that $\sigma = \text{Id}$ and that $b = 0$. We then have for a single example $x$:

$$z_{N+1} = Wz_N + Ux \tag{81}$$

$$z_N = W^N(z_0 + (I - W)^{-1}Ux) - (I - W)^{-1}Ux \tag{82}$$

With this trajectory, assuming $\ell(z) = \frac{1}{2}\|z - \omega\|_2^2$ the overall minimization problem then becomes:

$$\arg\min_{W,U} \frac{1}{2}\|W^N(z_0 + (I - W)^{-1}Ux) - (I - W)^{-1}Ux - \omega\|_2^2. \tag{83}$$

This problem is in general not even necessarily convex in $W$ and $U$.

## C  IMPLICIT DIFFERENTIATION PROOFS

**Lemma 1** (Convergence of the practical IFT gradient descent). *Under Assumption 3.1, Assumption 3.2 and Assumption 3.3, with $\ell$ strongly convex, $\exists N_0$ such that $\forall N > N_0$, $\exists \alpha_N > 0$ such that the sequences $\theta_{IFT}^{T,N}$ converge to a value denoted $\theta_{IFT}^{\star,N}$, dependent only on the initialization.*

*Proof.* Let us borrow the notations of Proposition 3. Let us denote $H = K_{\text{in}}^\top B$, $\bar{H} = BK_{\text{in}}^\top$ and $G = K_{\text{out}}^\top K_{\text{out}}$. We have:

$$p_N(\theta) = -(H^\dagger K_{\text{in}}^\top U)^\top \nabla l(z_N(\theta)) \tag{84}$$

$$= -(H^\dagger K_{\text{in}}^\top U)^\top (G z_N(\theta) - K_{\text{out}}^\top \omega) \tag{85}$$

$$= -(H^\dagger K_{\text{in}}^\top U)^\top (G(K_{\text{in}}^\top E_N U \theta + r_N) - K_{\text{out}}^\top \omega) \tag{86}$$

$$= -(H^\dagger K_{\text{in}}^\top U)^\top G K_{\text{in}}^\top E_N U \theta - (H^\dagger K_{\text{in}}^\top U)^\top (G r_N - K_{\text{out}}^\top \omega) \tag{87}$$

$$= -(\bar{H}^{-1} U)^\top K_{\text{in}} G K_{\text{in}}^\top E_N U \theta - (H^\dagger K_{\text{in}}^\top U)^\top (G r_N - K_{\text{out}}^\top \omega) \tag{88}$$

$$= X_N \theta - b_N \tag{89}$$

with $X_N = -(\bar{H}^{-1} U)^\top K_{\text{in}} G K_{\text{in}}^\top E_N U$ and $b_N = (H^\dagger K_{\text{in}}^\top U)^\top (G r_N - K_{\text{out}}^\top \omega)$. The affine dynamical system we need to study is therefore:

$$\theta_{\text{IFT}}^{T+1,N} = (I - \alpha_N X_N)\theta_{\text{IFT}}^{T,N} - \alpha_N b_N \tag{90}$$

If $X_N$ has only nonnegative eigenvalues real part, then we can use $\alpha_N = \frac{1}{\lambda_{\max} + \epsilon}$ where $\lambda_{\max}$ is the largest eigenvalue module of $X_N$ and $\epsilon > 0$. In this case the largest real part of an eigenvalue of $(I - \alpha_N X_N)$ is bounded in magnitude by 1 and the dynamical system converges.

We now need to show that $X_N$ has only nonnegative eigenvalues real part. To do so, let's write $X_N$ as the difference between a symmetric and a non-symmetric matrix, using $P_N(\bar{H}) = (I - \eta\bar{H})$:

$$X_N = -(\bar{H}^{-1} U)^\top K_{\text{in}} G K_{\text{in}}^\top E_N U \tag{91}$$

$$= -(\bar{H}^{-1} U)^\top K_{\text{in}} G K_{\text{in}}^\top (P_N(\bar{H}) - I)\bar{H}^{-1} U \tag{92}$$

$$= (\bar{H}^{-1} U)^\top K_{\text{in}} G K_{\text{in}}^\top \bar{H}^{-1} U - (\bar{H}^{-1} U)^\top K_{\text{in}} G K_{\text{in}}^\top P_N(\bar{H})\bar{H}^{-1} U \tag{93}$$

$$= X_{\text{sym}} - X_{N,\text{non-sym}} \tag{94}$$

If we take one unit eigenvector $v$ of $X_N$ with associated eigenvalue $\lambda$ we have:

$$\lambda = \|v\|^2 \lambda \tag{95}$$

$$= v^\top \lambda v \tag{96}$$

$$= v^\top (X_{\text{sym}} - X_{N,\text{non-sym}})v \tag{97}$$

$$= v^\top X_{\text{sym}} v - v^\top X_{N,\text{non-sym}} v \tag{98}$$

$$= \|K_{\text{out}} K_{\text{in}}^\top \bar{H}^{-1} U v\|_2^2 - v^\top (\bar{H}^{-1} U)^\top K_{\text{in}} G K_{\text{in}}^\top P_N(\bar{H})\bar{H}^{-1} U v \tag{99}$$

$$= \|K_{\text{out}} K_{\text{in}}^\top y\|_2^2 - y^\top K_{\text{in}} G K_{\text{in}}^\top P_N(\bar{H}) y \tag{100}$$

$$= \|K_{\text{out}} \gamma\|_2^2 - \gamma^\top K_{\text{out}}^\top K_{\text{out}} P_N(H)\gamma \tag{101}$$

with $y = \bar{H}^{-1} U$ and $\gamma = K_{\text{in}}^\top y$.

We can first notice that for $\gamma \in \ker(K_{\text{out}})$, $\lambda = 0$.

We now consider $\gamma \notin \ker(K_{\text{out}})$.

$$\gamma^\top K_{\text{out}}^\top K_{\text{out}} P_N(H)\gamma = \langle \gamma, P_N(H)\gamma \rangle_{K_{\text{out}}} \leq \|P_N(H)\|_{K_{\text{out}}} \|K_{\text{out}}\gamma\|_2^2 \tag{102}$$

Therefore, we have:

$$\lambda \geq (1 - \|P_N(H)\|_{K_{\text{out}}})\|K_{\text{out}}\gamma\|_2^2 \tag{103}$$

Because the inner procedure is a converging fixed-point iteration method, $|P_N(\lambda)|$ can be made arbitrarily small for all eigenvalues of $H$. In particular, $\exists N_0$ such that $\forall N > N_0$, $\|P_N(H)\|_{K_{\text{out}}} < 1$. Therefore, $\lambda \geq 0$.

This proof generalizes to gradient-based methods by replacing $P_N$ with the associated residual polynomial.

$\square$

**Theorem 2** (Equivalence of IFT and unrolled solutions for affine inner problems). *Under Assumption 3.1, Assumption 3.2 and Assumption 3.3, with $\ell$ strongly convex and $U$ surjective, we have:*

$$\theta_{IFT}^{\star,N} = \theta^{\star,N} \tag{14}$$

*Proof.* We borrow the notations from the proof of Proposition 3. Let us rewrite the root problem satisfied by $\theta_{\text{IFT}}^{\star,N}$:

$$(H^\dagger K_{\text{in}}^\top U)^\top \nabla l(z_N(\theta)) = 0 \tag{104}$$

$$\Leftrightarrow (H^\dagger K_{\text{in}}^\top U)^\top (Gz_N(\theta) - K_{\text{out}}^\top \omega) = 0 \tag{105}$$

$$\Leftrightarrow (H^\dagger K_{\text{in}}^\top U)^\top (G(K_{\text{in}}^\top E_N U\theta + r_N) - K_{\text{out}}^\top \omega) = 0 \tag{106}$$

$$\Leftrightarrow (H^\dagger K_{\text{in}}^\top U)^\top GK_{\text{in}}^\top E_N U\theta + (H^\dagger K_{\text{in}}^\top U)^\top (Gr_N - K_{\text{out}}^\top \omega) = 0 \tag{107}$$

$$\Leftrightarrow (H^\dagger K_{\text{in}}^\top U)^\top GK_{\text{in}}^\top E_N U\theta = -(H^\dagger K_{\text{in}}^\top U)^\top (Gr_N - K_{\text{out}}^\top \omega) \tag{108}$$

$$\Leftrightarrow (H^\dagger K_{\text{in}}^\top U)^\top GK_{\text{in}}^\top E_N U\theta = -(H^\dagger K_{\text{in}}^\top U)^\top K_{\text{out}}^\top (K_{\text{out}} r_N - \omega) \tag{109}$$

$$\Leftrightarrow (H^\dagger K_{\text{in}}^\top U)^\top GK_{\text{in}}^\top E_N U\theta = -(K_{\text{out}} H^\dagger K_{\text{in}}^\top U)^\top (K_{\text{out}} r_N - \omega) \tag{110}$$

$$\Leftrightarrow (K_{\text{in}}^\top \bar{H}^{-1} U)^\top GK_{\text{in}}^\top E_N U\theta = -(K_{\text{out}} K_{\text{in}}^\top \bar{H}^{-1} U)^\top (K_{\text{out}} r_N - \omega) \tag{111}$$

$$\Leftrightarrow (K_{\text{out}} K_{\text{in}}^\top \bar{H}^{-1} U)^\top K_{\text{out}} K_{\text{in}}^\top E_N U\theta = -(K_{\text{out}} K_{\text{in}}^\top \bar{H}^{-1} U)^\top (K_{\text{out}} r_N - \omega) \tag{112}$$

$$\Leftrightarrow K_{\text{out}} K_{\text{in}}^\top E_N U\theta = -(K_{\text{out}} r_N - \omega) + \theta_{\ker((K_{\text{out}} K_{\text{in}}^\top \bar{H}^{-1} U)^\top)} \tag{113}$$

$$\Leftrightarrow K_{\text{out}} K_{\text{in}}^\top E_N U\theta = -\mathcal{P}(K_{\text{out}} A_N)(K_{\text{out}} r_N - \omega) + \mathcal{P}(K_{\text{out}} A_N)\theta_{\ker((K_{\text{out}} K_{\text{in}}^\top \bar{H}^{-1} U)^\top)} \tag{114}$$

$$\Leftrightarrow K_{\text{out}} K_{\text{in}}^\top E_N U\theta = -\mathcal{P}(K_{\text{out}} A_N)(K_{\text{out}} r_N - \omega) + \mathcal{P}(K_{\text{out}} A_N)\theta_{\text{Im}(K_{\text{out}} K_{\text{in}}^\top \bar{H}^{-1} U)^\perp} \tag{115}$$

$$\Leftrightarrow K_{\text{out}} K_{\text{in}}^\top E_N U\theta = -\mathcal{P}(K_{\text{out}} K_{\text{in}}^\top)(K_{\text{out}} r_N - \omega) + \mathcal{P}(K_{\text{out}} K_{\text{in}}^\top)\theta_{\text{Im}(K_{\text{out}} K_{\text{in}}^\top)^\perp} \tag{116}$$

$$\Leftrightarrow K_{\text{out}} K_{\text{in}}^\top E_N U\theta = -\mathcal{P}(K_{\text{out}} K_{\text{in}}^\top)(K_{\text{out}} r_N - \omega) \tag{117}$$

$$\Leftrightarrow U\theta = -E_N^{-1}(K_{\text{out}} K_{\text{in}}^\top)^\dagger \mathcal{P}(K_{\text{out}} A_N)(K_{\text{out}} r_N - \omega) + E_N^{-1}\theta_{\ker(K_{\text{out}} K_{\text{in}}^\top)} \tag{118}$$

And we end up with the same characterization as (43), which concludes the proof.

$\square$

## D  GENERALIZATION

In order to get a more consistent result, we should make sure that it holds in a generalization setting, i.e. on samples not included in $\mathcal{D}_{\text{train}}$. To do so, we can assume that each sample in $\mathcal{D}_{\text{train}}$ has its own separate inner procedure and that the outer loss is the sum of all separate outer losses. Using the stacked notation, i.e. $\mathbf{x} = [x^{(1)}, \dots, x^{(n)}]^\top$, $\mathbf{z}_N(\theta, \mathbf{x}) = [z_N(\theta, x^{(1)}), \dots, z_N(\theta, x^{(n)})]^\top$, $\mathbf{f}(\mathbf{z}, \theta, \mathbf{x}) = [f(z^{(1)}, \theta, x^{(1)}), \dots, f(z^{(n)}, \theta, x^{(n)})]^\top$ and $\ell(\mathbf{z}, \mathbf{x}) = \frac{1}{n}\sum_i \ell(z^{(i)}, x^{(i)})$, we can write the following optimization problem:

$$\arg\min_\theta \ell(\mathbf{z}^\star(\theta, \mathbf{x}), \mathbf{x}) = \frac{1}{n}\sum_i \ell(z^\star(\theta, x^{(i)}), x^{(i)})$$
$$\text{s.t.} \quad \mathbf{f}(\mathbf{z}, \theta, \mathbf{x}) = \begin{bmatrix} f(z^\star(\theta, x^{(1)}), \theta, x^{(1)}) \\ \vdots \\ f(z^\star(\theta, x^{(n)}), \theta, x^{(n)}) \end{bmatrix} = 0 \tag{119}$$

with:

$$\ell(z, x) = \frac{1}{2}\|K_{\text{out}}(x)z - \omega(x)\|_2^2$$
$$f(z, \theta, x) = K_{\text{in}}(x)^\top (B(x)z + U(x)\theta + c(x)) \tag{120}$$

If we stack the individual matrices and vectors as follows, we get the same bound as Theorem 1:

$$\mathbf{U} = \begin{bmatrix} U(x^{(1)}) \\ \vdots \\ U(x^{(n)}) \end{bmatrix} \tag{121}$$

$$\mathbf{B} = \begin{bmatrix} B(x^{(1)}) & & \\ & \ddots & \\ & & B(x^{(n)}) \end{bmatrix} \tag{122}$$

$$\mathbf{c} = \begin{bmatrix} c(x^{(1)}) \\ \vdots \\ c(x^{(n)}) \end{bmatrix} \tag{123}$$

$$\mathbf{K}_{\text{in}} = \begin{bmatrix} K_{\text{in}}(x^{(1)}) & & \\ & \ddots & \\ & & K_{\text{in}}(x^{(n)}) \end{bmatrix} \tag{124}$$

$$\mathbf{K}_{\text{out}} = \frac{1}{\sqrt{n}} \begin{bmatrix} K_{\text{out}}(x^{(1)}) & & \\ & \ddots & \\ & & K_{\text{out}}(x^{(n)}) \end{bmatrix} \tag{125}$$

$$\omega = \frac{1}{\sqrt{n}} \begin{bmatrix} \omega(x^{(1)}) \\ \vdots \\ \omega(x^{(n)}) \end{bmatrix} \tag{126}$$

That is we have:

$$\frac{1}{n}\sum_i \ell(z_{N+\Delta N}(\theta, x^{(i)}), x^{(i)}) - \frac{1}{n}\sum_i \ell(z_N(\theta, x^{(i)}), x^{(i)}) \tag{127}$$

$$\geq -\frac{1}{2}\| \left( \mathcal{P}(\mathbf{K}_{\text{out}}\mathbf{K}_{\text{in}}^\top) - \mathcal{P}(\mathbf{K}_{\text{out}}\mathbf{K}_{\text{in}}^\top \mathbf{E}_N \mathbf{U}) \right) (\mathbf{K}_{\text{out}}\mathbf{r}_N - \omega)\|_2^2 \tag{128}$$

where:

$$\mathbf{E}_N = \begin{bmatrix} E_N(x^{(1)}) & & \\ & \ddots & \\ & & E_N(x^{(n)}) \end{bmatrix} \tag{129}$$

$$E_N(x^{(i)}) = \left( (I - \eta B(x^{(i)}) K_{\text{in}}(x^{(i)})^\top)^N - I \right) (B(x^{(i)}) K_{\text{in}}(x^{(i)})^\top)^{-1} \tag{130}$$

$$\mathbf{r}_N = \mathbf{K}_{\text{in}}^\top \mathbf{E}_N (\mathbf{c} + \mathbf{B}\mathbf{z}_0) + \mathbf{z}_0 \tag{131}$$

For simplicity, we denote the bound as:

$$\mathcal{B}_N(\mathcal{D}_{\text{train}}) = -\frac{1}{2}\| \left( \mathcal{P}(\mathbf{K}_{\text{out}}\mathbf{K}_{\text{in}}^\top) - \mathcal{P}(\mathbf{K}_{\text{out}}\mathbf{K}_{\text{in}}^\top \mathbf{E}_N \mathbf{U}) \right) (\mathbf{K}_{\text{out}}\mathbf{r}_N - \omega)\|_2^2 \tag{132}$$

One can wonder what is the impact increasing the size of the training dataset $\mathcal{D}_{\text{train}}$ on $\mathcal{B}_N(\mathcal{D}_{\text{train}})$ for a fixed model. If the different functions $U, B, K_{\text{in}}, \ldots$ are sufficiently smooth and the data support is compact, then $\mathcal{B}_N(\mathcal{D}_{\text{train}})$ is bounded, i.e. $\lim_{n \to +\infty} -\mathcal{B}_N(\mathcal{D}_{\text{train}}) < +\infty$.
This can be seen with the following rewriting of $\mathcal{B}_N(\mathcal{D}_{\text{train}})$:

$$\mathcal{B}_N(\mathcal{D}_{\text{train}}) = -\frac{1}{2n}\sum_i \|\mathcal{P}(K_{\text{out}}^{(i)} K_{\text{in}}^{(i),\top})(K_{\text{out}}^{(i)} r_N^{(i)} - \omega^{(i)}) + K_{\text{out}}^{(i)} K_{\text{in}}^{(i),\top} E_N^{(i)} U^{(i)} \theta^{\star,N}\| \tag{133}$$

where we replaced the evaluation on $x^{(i)}$ with a superscript $^{(i)}$ for conciseness.

Now, in the case where the different functions $U, B, K_{\text{in}}, \ldots$ are constant with $U(x)$ surjective then we find ourselves in the overparametrization case and it is $\mathcal{B}_N(\mathcal{D}_{\text{train}}) = 0$. If we are sufficiently close to constant functions, i.e. if they are sufficiently smooth, the lower bound remains meaningful. For generalization, we have the following result:

**Corollary D.1.** *Assuming that $\ell$ and $f$ have expressions following (120) and are smooth, that the inner procedure is a converging fixed-point iteration method, that $\theta^{\star,N}$ is obtained via gradient descent, and that the distribution $\Pi$ from which we sample $\mathcal{D}_{train}$ and $x$ has a compact support $\mathcal{X}$, then with probability $1 - \delta$, we have:*

$$\mathbb{E}_{x \sim \Pi} \left[ \ell(z_{N+\Delta N}(\theta^{\star,N}, x), x) - \ell(z_N(\theta^{\star,N}, x), x) \right] \geq \mathcal{B}_N(\mathcal{D}_{train}) - \frac{D_\delta(\log(|\mathcal{D}_{train}|)}{\sqrt{|\mathcal{D}_{train}|}}, \quad (134)$$

*where $D_\delta$ is a monotone increasing function depending on $\delta$, the smoothness of $f$ and $\ell$ w.r.t $x$ and the size of $\mathcal{X}$.*

The proof relies on expressing the problem as an estimation error problem and borrowing tools from the probably approximately correct learning theory (McDiarmid's inequality and Rademacher's complexity).

*Proof.* From Theorem 1, we have that:

$$\frac{1}{n} \sum_i \ell(z_{N+\Delta N}(\theta, x^{(i)}), x^{(i)}) - \frac{1}{n} \sum_i \ell(z_N(\theta, x^{(i)}), x^{(i)}) \geq \mathcal{B}_N(\mathcal{D}_{\text{train}}) \quad (135)$$

We can then control the following difference, for $P = N + \Delta N$:

$$\left| \frac{1}{n} \sum_i \ell(z_P(\theta^{\star,N}, x^{(i)}), x^{(i)})) - \mathbb{E}_x \left[ \ell(z_P(\theta^{\star,N}, x), x) \right] \right|. \quad (136)$$

In order to use the PAC framework, we need the quantity in the expectation to be independent of the dataset $\mathcal{D}_{\text{train}}$, which is not the case in (136) because of the dependency via $\theta^{\star,N}$. In order to remove this dependency we will introduce the following set:

$$\Theta_{N,n} = \{ \text{GD} \left( \theta \mapsto \ell(z_N(\theta, \mathcal{D}_n), \mathcal{D}_n) \right) \in \mathbb{R}^{d_\theta} | \mathcal{D}_n \in \mathcal{X}^n \} \quad (137)$$

In plain English, $\Theta_{N,n}$ is the set of minimizers of (P) obtained via gradient descent for all possible datasets $\mathcal{D}_n$ of size $n$ drawn from the distribution $\Pi$. To define this set, we introduced GD the function that computes the limit of the gradient descent for a given function. By definition of the supremum, we have:

$$\left| \frac{1}{n} \sum_i \ell(z_P(\theta^{\star,N}, x^{(i)})), x^{(i)})) - \mathbb{E}_x \left[ \ell(z_P(\theta^{\star,N}, x), x) \right] \right| \quad (138)$$

$$\leq \sup_{\theta \in \Theta_{N,n}} \left| \frac{1}{n} \sum_i \ell(z_P(\theta, x^{(i)})), x^{(i)})) - \mathbb{E}_x \left[ \ell(z_P(\theta, x), x) \right] \right|. \quad (139)$$

As classically termed in the PAC framework, we define $\Delta(\mathcal{D}_{\text{train}})$, the representativeness of $\mathcal{D}_{\text{train}}$, as:

$$\Delta(\mathcal{D}_{\text{train}}) = \sup_{\theta \in \Theta_{N,n}} \left| \frac{1}{n} \sum_i \ell(z_P(\theta, x^{(i)})), x^{(i)})) - \mathbb{E}_x \left[ \ell(z_P(\theta, x), x) \right] \right|. \quad (140)$$

Using McDiarmid's inequality we then have the following inequality with probability $1 - \delta$:

$$\left| \Delta(\mathcal{D}) - \mathbb{E}_\mathcal{D} \left[ \Delta(\mathcal{D}) \right] \right| \leq Q \sqrt{\frac{2 \log(1/\delta)}{|\mathcal{D}|}}, \quad (141)$$

with $Q$ an upper bound on $h_P(\theta, x) = \ell(z_P(\theta, x), x) \; \forall \theta \in \Theta_{N,n}$ and $\forall x \in \mathcal{X}$.

We now turn to show that the upper bound $Q$ exists. From (46), we have $h_P(\theta, x) = \frac{1}{2} \| C(x) E_P(x) U(x) \theta^{\star,N} + K_{\text{out}}(x) r_P(x) - \omega(x) \|_2^2$ extending the notations of the proof of Theorem 1. Since $\mathcal{X}$ is a compact space and $h$ is smooth in $\theta$ and $x$, we need to prove that $\Theta_{N,n}$ is a compact space to show that the upper bound exists.

$\Theta_{N,n}$ is the image of $\mathcal{X}^n$, a compact space, by the function $\rho : \mathcal{D}_n \mapsto \text{GD} \left( \theta \mapsto \ell(z_N(\theta, \mathcal{D}_n), \mathcal{D}_n) \right)$. We first consider the function $\phi$ that maps from a dataset $\mathcal{D}_{\text{train}}$ to the sets of parameters of the quadratic $(\mathbf{C}(x)\mathbf{E}_P(x)\mathbf{U}(x))_{x \in \mathcal{D}_{\text{train}}}$ and $(\mathbf{K}_{\text{out}}(x)\mathbf{r}_P(x) - \omega(x))_{x \in \mathcal{D}_{\text{train}}}$. By hypothesis on the mapping for these parameters, $\phi$ is smooth and $\phi(\mathcal{D}_{\text{train}})$ is a compact. Then, we introduce $\lambda : (\Gamma, \alpha) \mapsto \text{GD}(\theta \mapsto \frac{1}{2} \| \Gamma \theta + \alpha \|_2^2) = -(\Gamma^\top \Gamma)^{-1} \Gamma^\top \alpha$ which is smooth. Finally, we have that $\rho = \lambda \circ \phi$ and so is smooth

as a composition of smooth functions. We can conclude that $\Theta_{N,n}$ is compact and that $Q$ indeed exists.

We can then use Rademacher's complexity $\mathcal{R}_{|\mathcal{D}|}$ (see Mohri et al. 2018 for a definition) to bound $\mathbb{E}_{\mathcal{D}}\left[\Delta(\mathcal{D})\right]$. First, we use the symmetrization lemma (Mohri et al., 2018, Theorem 3.3) to get:

$$\mathbb{E}_{\mathcal{D}}\left[\Delta(\mathcal{D})\right] \leq 2\mathcal{R}_{|\mathcal{D}|}(\mathcal{H}), \tag{142}$$

with $\mathcal{H} = \{h_P(\theta, \cdot), \theta \in \Theta_{N,n}\}$. Since $\Pi$ has a compact support $\mathcal{X}$, there exists a max norm of $A(x) = K_{\text{out}}(x)K_{\text{in}}^{\top}(x)E_N(x)U(x)$ that we denote $A_{\infty}$. This means that we have the following $\forall x \in \mathcal{X}$ and $\forall \theta, \theta' \in \Theta_{N,n}$:

$$|h_P(x, \theta) - h_P(x', \theta')| \leq 2TA_{\infty}^2 \|\theta - \theta'\|_2 \tag{143}$$

where $T$ is the minimum radius of a norm-2 ball that includes $\Theta_{N,n}$ (possible because it is compact). This means that the parametrization of $\mathcal{H}$ is $2TA_{\infty}^2$-Lipschitz w.r.t the Euclidean distance on $\Theta_{N,n}$ and we can use the smoothly parametrized class theorem of Bartlett (2013, Lecture 13) and get:

$$\mathcal{R}_n(\mathcal{H}) \leq 2\kappa T^2 A_{\infty}^2 \sqrt{\frac{d_\theta \log(2T^2 A_{\infty}^2 n)}{n}}, \tag{144}$$

where $\kappa$ is a constant. To conclude, with $L = 2T^2 A_{\infty}^2$, we have:

$$\mathbb{E}_{\mathcal{D}}\left[\Delta(\mathcal{D})\right] \leq 2\kappa L \sqrt{\frac{d_\theta \log(Ln)}{n}} \tag{145}$$

$$\implies \Delta(\mathcal{D}) \leq 2\kappa L \sqrt{\frac{d_\theta \log(Ln)}{n}} + Q\sqrt{\frac{2\log(1/\delta)}{|\mathcal{D}|}} \tag{146}$$

$$\equiv \Delta(\mathcal{D}) \leq \frac{2\kappa L \sqrt{d_\theta \log(L|\mathcal{D}|)} + \sqrt{2\log(1/\delta)}}{\sqrt{|\mathcal{D}|}} \tag{147}$$

$$\implies \left| \frac{1}{n} \sum_i \ell(z_P(\theta^{\star,N}, x_i), x_i) - \mathbb{E}_x\left[\ell(z_P(\theta^{\star,N}, x), x)\right] \right| \leq \frac{2\kappa L \sqrt{d_\theta \log(L|\mathcal{D}|)} + \sqrt{2\log(1/\delta)}}{\sqrt{|\mathcal{D}|}} \tag{148}$$

$$\implies \mathbb{E}_{x\sim\Pi}\left[\ell(z_{N+\Delta N}(\theta^{\star,N}, x), x)\right] - \mathbb{E}_{x\sim\Pi}\left[\ell(z_N(\theta^{\star,N}, x), x)\right] \geq \mathcal{B}_N(\mathcal{D}_{\text{train}}) - \frac{D_\delta(\log|\mathcal{D}_{\text{train}}|)}{\sqrt{|\mathcal{D}_{\text{train}}|}} \tag{149}$$

$$\equiv \mathbb{E}_{x\sim\Pi}\left[\ell(z_{N+\Delta N}(\theta^{\star,N}, x), x) - \ell(z_N(\theta^{\star,N}, x), x)\right] \geq \mathcal{B}_N(\mathcal{D}_{\text{train}}) - \frac{D_\delta(\log|\mathcal{D}_{\text{train}}|)}{\sqrt{|\mathcal{D}_{\text{train}}|}}, \tag{150}$$

with $D_\delta(\log(n)) = 2\left(2\kappa L\sqrt{d_\theta}\sqrt{\log(L) + \log(n)} + \sqrt{2\log(1/\delta)}\right)$. $\qquad\square$

## E  EXTENDED RELATED WORKS

**Theoretical analysis of implicit deep learning practice**   Relatively few works have looked at how practical implicit deep learning implementations affect the final solution. Most notable is the one of Vicol et al. (2022). In this work, the authors looked at the implicit biases caused by warm-starting in the inner optimization problem and the use of approximate hypergradients on the final solution. They proved theoretical results in a quadratic bilevel optimization setting, and showed empirical results on dataset distillation and data augmentation network learning. Their conclusion is that warm-starting in the inner problem leads to overfitting and information leakage, and that using a higher quality hypergradient leads to min-norm solutions for the outer problem. Our work is complementary to theirs in that they have not considered non warm-start cases with a fixed number of iterations. We also highlight that the notion of overparametrization later introduced in our work is different from theirs. Their notion refers to the inner or outer problem having potentially more than one solution (a setup we cover).

**Convergence results in DEQs** Some works have also tackled the notion of convergence in DEQs. Winston & Kolter (2020) and Feng et al. (2023) tackled the problem of understanding what conditions are needed for the inner problem of DEQs to be well-defined in the sense that it indeed converges to a unique solution. Ling et al. (2023) looked at the convergence of gradient descent for the outer problem with a quadratic outer loss. Interestingly, they also uncovered a link with overparametrization. We want to highlight that these results differ from Lemma 1 in that we are really interested in a non-asymptotic outer convergence, whereas the previously mentioned papers only tackle cases where the inner problem is solved exactly.

## F GRADIENT-BASED METHODS AND RESIDUAL POLYNOMIALS

Using the notations of Pedregosa (2020), a gradient-based method can be defined as having iterates of the following form:

$$z_{N+1} = z_N + \sum_{i=0}^{N-1} c_i^{(N)}(z_{i+1} - z_i) + c_i^{(N)} \nabla f(z_N), \tag{151}$$

The residual polynomials of this gradient-based method are then defined recursively as:

$$P_{N+1}(\lambda) = (1 + c_N^{(N)}\lambda)P_N(\lambda) + \sum_{i=0}^{N-1} c_i^{(N)}(P_{i+1}(\lambda) - P_i(\lambda)) \tag{152}$$
$$P_0(\lambda) = 1$$

**Lemma F.1.** *Let* $F : \mathbb{R}^{d_z} \to \mathbb{R}$, $F(z) = \frac{1}{2}\|Kz + c\|_2^2$. *We define* $z_N$ *as the* $N$-*th iterate of a gradient-based method with associated residual polynomial* $P_N$ *(Hestenes et al., 1952; Fischer, 2011) and initial condition* $z_0$. *Then the closed form expression of* $z_N$ *is:*

$$z_N = P_N(K^\top K)z_0 + (P_N(K^\top K) - I)(K^\top K)^\dagger c \tag{153}$$

*Proof.* Writing $H = K^\top K$ the hessian of $F$ and $z^\star \in \arg\min_z F(z)$, we have the following equality (Hestenes et al., 1952; Fischer, 2011; Pedregosa, 2020):

$$z_N - z^\star = P_N(H)(z_0 - z^\star) \tag{154}$$

Rewriting it, leads to:

$$z_N = P_N(H)z_0 - (P_N(H) - I)z^\star \tag{155}$$

And we have that $z^\star$ is such that $\nabla F(z^\star) = Hz^\star + K^\top c = 0$. We can take for example $z^\star = -H^\dagger K^\top c$. Therefore:

$$z_N = P_N(H)z_0 + (P_N(H) - I)H^\dagger K^\top c \tag{156}$$

$\square$

**Proposition 3.** *Let us assume that* $f$ *is the gradient in* $z$ *of a function* $F$ *quadratic in* $z$ *and linear in* $\theta$, *convex and bounded from below. Then any converging gradient-based method minimizing* $F$ *is a time-invertible linear procedure.*

*Proof.* Let us give the expression of $F$:

$$F(z, \theta) = \frac{1}{2}\|K_{\text{in}}z + U\theta + c\|_2^2, \tag{157}$$

with $K_{\text{in}} \in \mathbb{R}^{d_x \times d_z}$ surjective (if not we can always reformulate $F$ to have it surjective), $U \in \mathbb{R}^{d_x \times d_\theta}$, $c \in \mathbb{R}^{d_z}$. From Lemma F.1, writing $K_{\text{in}}^\top K_{\text{in}} = H$, we know that $z_N = P_N(H)z_0 + (P_N(H) - I)H^\dagger(K_{\text{in}}^\top(U\theta + c))$. Rewriting this, with $\bar{H} = K_{\text{in}}K_{\text{in}}^\top$:

$$z_N = P_N(H)z_0 + (P_N(H) - I)H^\dagger K_{\text{in}}^\top(U\theta + c) \tag{158}$$
$$= (P_N(H) - I)H^\dagger K_{\text{in}}^\top U\theta + P_N(H)z_0 + (P_N(H) - I)H^\dagger K_{\text{in}}^\top c \tag{159}$$
$$= K_{\text{in}}^\top(P_N(\bar{H}) - I)\bar{H}^\dagger U\theta + P_N(H)z_0 + (P_N(H) - I)H^\dagger K_{\text{in}}^\top c \tag{160}$$
$$= K_{\text{in}}^\top E_N U\theta + r_N \tag{161}$$

with $E_N = (P_N(\bar{H}) - I)\bar{H}^\dagger$ and $r_N = P_N(H)z_0 + (P_N(H) - I)H^\dagger K_{\text{in}}^\top c$. Because $K_{\text{in}}$ is surjective $\bar{H}$ is invertible and $\bar{H}^\dagger = \bar{H}^{-1}$ is as well. Because $P_N$ is associated with a converging gradient-based method (see more in Appendix F), $\exists N_0$ such that $\forall N > N_0, P_N(\lambda) \neq 1, \forall \lambda \in [\lambda_{\min}; \lambda_{\max}]$, $(P_N(\bar{H}) - I)$ is invertible. Therefore, $E_N$ is invertible as the product of 2 invertible matrices. This concludes the proof. $\square$

Table 1: Number of iterations for fixed point resolution during training for pretrained DEQs for the different tasks covered.

| Task | $N$ |
|---|---|
| Image classification | 26 |
| Image segmentation | 27 |
| Language modeling | 30 |
| Optical Flow estimation | 36 |
| Single-image super-resolution | 100 |
| Landmark detection | 5 |

*Remark* F.2. Gradient-based methods have a rate of convergence proportional to $\max_{\lambda \in [0, \lambda_{max}]} |P_N(\lambda)|$ (Pedregosa, 2020) where $P_N$ is their associated residual polynomial. Therefore, for any converging gradient-based method, $\exists N_0$ s.t. $\forall N > N_0, \forall \lambda \in [\lambda_{min}; \lambda_{max}] |P_N(\lambda)| < 1$.

*Remark* F.3. For gradient descent with step size $\frac{1+\epsilon}{\lambda_{max}}$ with $-1 < \epsilon < 1$, we have $P_N(\lambda) = (1 - \frac{1+\epsilon}{\lambda_{max}}\lambda)^N$ (Pedregosa, 2020) $\forall N > 0$, which means that $P_N(\lambda) \neq 1$ if $\lambda \in [\lambda_{min}; \lambda_{max}]$. In the case of gradient descent with an appropriate step size, the inequality (9) is therefore always true for all optimization steps.

*Remark* F.4. For gradient descent with momentum with admissible parameters as defined by Pedregosa (2020)[Blog 2], we can reuse computations made to check the convergence to get $N_0$ from Remark F.3. For momentum $m$, $N_0$ is such that:

$$m^{\frac{N_0}{2}}(1 + \frac{1-m}{1+mN_0}) < 1 \tag{162}$$

## G    EXPERIMENTAL DETAILS

### G.1    DEQs

In all DEQs experiments, except the stability experiment, we reuse the data, architecture, code (in PyTorch (Paszke et al., 2019)) and weights of the original works. The only difference with the original works is that we use a different number of inner steps for the fixed point resolution, and set the tolerance to a value sufficiently low, so that the maximum number of inner steps is always reached ($10^{-7}$ generally). We list here the links to the original works public GitHub repositories which contain information on how to download the data, the weights and how to perform inference:

- Image classification on ImageNet (Deng et al., 2009) and CIFAR (Krizhevsky, 2009) and Image segmentation on Cityscapes (Cordts et al., 2016): locuslab/deq/MDEQ-Vision (Bai et al., 2020)
- Language modeling on WikiText (Merity et al., 2017): locuslab/deq/DEQ-Sequence (Bai et al., 2019)
- Optical Flow Estimation on Sintel (Butler et al., 2012): locuslab/deq-flow (Bai et al., 2022a)
- Single-image Super resolution on CBSD68 (Martin et al., 2001): wustl-cig/ELDER (Zou et al., 2023)
- Landmark detection on WFLW-V (Micaelli et al., 2023): polo5/LDEQ_RwR (Micaelli et al., 2023)

**How do we know that the maximum number of iterations is always reached rather than the tolerance?**    While the original DEQ implementation (`github.com/locuslab/deq`) does feature a mechanism to stop the fixed point resolution (generally done via the Broyden method) when a tolerance is reached, it is never used in practice. We can observe this via 2 different methods:

- Logging of the iteration count: it is always the maximum number of iterations specified.
- Comparison of the specified tolerance and the actual error reached: as can be seen in Figure 2 the error reached for ImageNet for example is about $0.2$ while the tolerance set in training is of the order of $10^{-3}$.

Finally, Bai et al. (2020, Section B.2.) mention this aspect in their Multiscale DEQ work: "In practice, we stop the Broyden iterations at some threshold limit (e.g., 22 iterations)".

## G.2 DEQs stability

In order to evaluate the stability of DEQs trained with unrolling (i.e. backpropagating through the iterates of Broyden's method), we needed to implement a differentiable version of Broyden's method. Except for this, the training code and data pipelines are taken from the original work (Bai et al., 2020) for image classification on CIFAR-10 (Krizhevsky, 2009). We simply vary the number of inner steps used for the fixed point resolution and the tolerance of the fixed point resolution at test-time. The networks used were those following the `TINY` configuration (see the original configuration file for more details).

One might notice that Broyden's method is usually not differentiable because of the use of a line search. As for the typical DEQ setting, we did not use a line search to train DEQs, whether unrolled or not, and just kept a fixed step size of 1, making it differentiable.

## G.3 (I)MAML

We recall the main difference between MAML and iMAML. In MAML, the meta parameters are used as an initialization to a gradient descent for task adapted networks, while in iMAML, the meta parameters are used as an anchor point for the task adapted weights. Formally, for iMAML, the inner loss is modified to include a regularization term that penalizes the $\ell_2$-norm of the difference between the task adapted parameters and the meta-parameters. Thanks to this formulation, iMAML can use implicit differentiation to compute the hypergradient, while MAML has to rely on unrolling.

We reimplemented the (i)MAML framework in Jax (Bradbury et al., 2018) using the recently developed Jaxopt library (Blondel et al., 2022). The reason for this was that we wanted a faster implementation, made possible by the machinery of Jax and Jaxopt together.

For the sinusoid regression task, we used the same architecture and hyperparameters as the one introduced by Finn et al. (2017) (respectively the same hyperparameters as the one of Rajeswaran et al. (2019), including the use of line search for gradient descent). Each sinusoid was generated on the fly, both for test and train data.

The architecture was a 2-hidden-layer MLP with 40 hidden units per layer, transductive batch normalization (i.e. the batch statistics are not stored) and ReLU activations. For MAML, the inner gradient descent had a step size of 0.01. For iMAML, the inner gradient descent use line search for a maximum of 16 iterations. The outer optimization was carried out for 70k steps using Adam (Kingma & Ba, 2015) with a learning rate of $10^{-3}$ and all other hyperparameters set to their default values, the meta batch size was 25 and 10 samples were used for validation (outer) loss computation and for the inner loss definition. The samples are generated on-the-fly.

## G.4 Compute

All the experiments except the theorem illustrations were run on a public HPC cluster, providing NVIDIA V100 GPUs. According to the numbers provided by this public HPC cluster, a maximum of 220 GPU days (5291 GPU hours) were used for the experiments (conservative upper bound), which includes all the reruns due to bugs, the experiments not reported in this paper and potential other projects worked on at the same time.

# H Additional results

## H.1 Inner iterations Overfitting for DEQs on training data

While in Figure 2 the reported performance is the test set performance, the theory developed in Section 3 concerns only the training set performance. In order to make sure that the behavior we are noticing on the test set performance is not due to some effect of lack of generalization we also report the same figure for ImageNet on training data in Figure H.1.

## H.2 Inner iterations Overfitting for DEQs on training loss

While in Figure 2 the reported performance is not the training loss but for example for image classification the error, the theory developed in Section 3 concerns only the training loss. In order to make sure that the behavior we are noticing is not due to a discrepancy between the optimized loss and the performance we also report the same figure for ImageNet on training loss in Figure H.2.

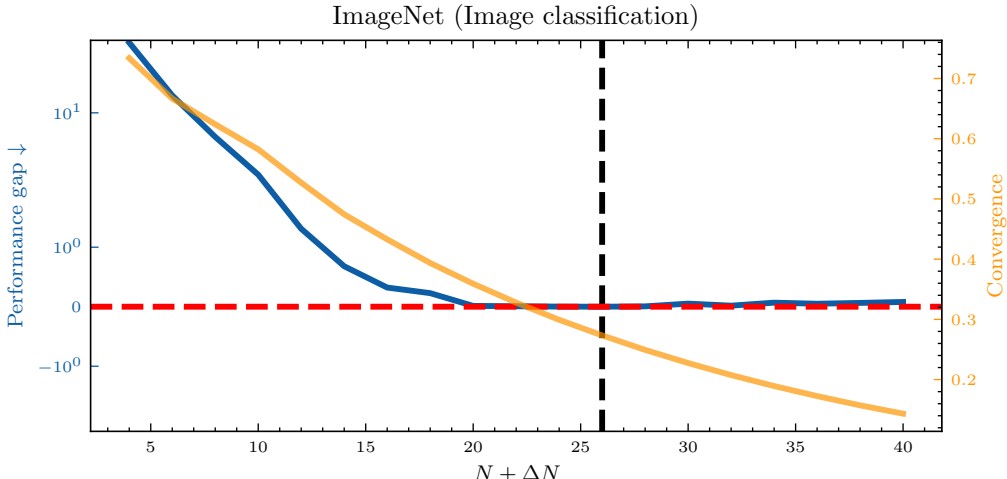

Figure H.1: **Inner iterations overfitting for DEQs on train data.** We report the training set error for different inner optimization times.

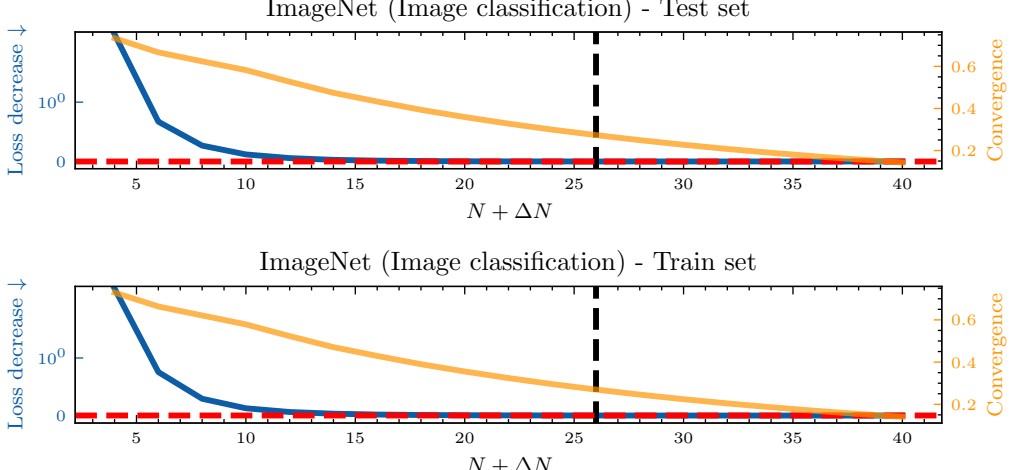

Figure H.2: **Inner iterations overfitting for DEQs on the training loss.** We report the training set error for different inner optimization times.

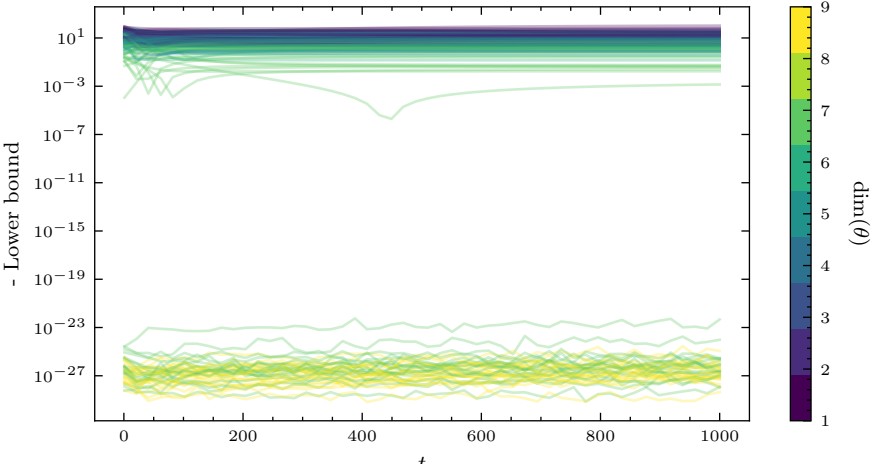

Figure H.3: **Impact of inner problem underparametrization for non strongly convex cases.** Negative Lower bound, i.e. $\frac{1}{2}\|(\mathcal{P}(K_{\text{out}}K_{\text{in}}) - \mathcal{P}(K_{\text{out}}K_{\text{in}}E_N U))(K_{\text{out}}r_N - z^\star)\|_2^2$, from Theorem 1. The inner and outer problem are not strongly convex, and the dimension of $z$ is 10. Roughly speaking, the inner problem would therefore be fully parameterized in $\theta$ if it was in dimension 10. We compute the lower bound for different inner optimization times and 20 seeds.

### H.3 LOWER BOUND FOR NON OVERPARAMETRIZED INNER PROCEDURES BUT WITH NON STRONGLY CONVEX INNER AND OUTER PROBLEMS

We see in Figure H.3 that the lower bound can reach 0 for a much smaller $\theta$ dimension when the inner and outer problem are not strongly convex than when they are.

## I LINK BETWEEN LINEAR AND NON-LINEAR IMPLICIT MODELS

We consider the implicit problem

$$\min \ell(z^*(\theta)) \text{ such that } f(z^*(\theta), \theta) = 0 \ , \tag{163}$$

where $\ell$ and $f$ are smooth functions, but contrary to the analysis in Section 3, $\ell$ is not assumed to be a quadratic function, and $f$ is not assumed to be affine. Hereafter, we derive an approximation theory between the solutions to the problem (163) and the linear problem that corresponds to a linear expansion of $f$ around the solution.

More precisely, we let $\theta^*$ the solution to (163), write for short $z^* = z^*(\theta)$, and define a linearized version of $f$ around $(z^*, \theta^*)$ as the Taylor expansion

$$g(z, \theta) = A(z - z^*) + B(\theta - \theta^*) \text{ with } A = \frac{\partial f}{\partial z}(z^*, \theta^*) \text{ and } B = \frac{\partial f}{\partial \theta}(z^*, \theta^*) \tag{164}$$

The function $g$ is now an affine function, exactly like the functions we consider in Section 3, eq. (7). We make the following assumption on the error.

**Assumption I.1** (Quadratic error). We define $R(z, \theta) = f(z, \theta) - g(z, \theta)$. There exists $\alpha, \beta > 0$ such that for all $\theta, z$, it holds

$$\|R(z, \theta)\| \leq \alpha \|z - z^*\|^2 + \beta \|\theta - \theta^*\|^2 \ .$$

This assumption is very weak; it is, for instance, verified if $f$ is twice differentiable and has at most quadratic growth, or if the functions are defined over a compact. We define $\phi_f(z_0, \theta, n)$ the output of $n$ iterations of the power method on $f$ with step-size 1 starting from $z_0$, i.e. $\phi_f(z_0, \theta, n) = z_n$ where $z_n$ is defined recursively by

$$z_{n+1} = z_n - f(z_n, \theta). \tag{165}$$

We assume that these iterations converge to $z^*(\theta)$ as $n$ goes to infinity:

**Assumption I.2** (Convergence of the power method). $\|I - A\|_2 = \lambda < 1$.

We define similarly the function $\phi_g(z_0, \theta, n)$ the output of $n$ iterations of the power method on $g$ with step-size 1 starting from $z_0$, i.e. $\phi_g(z_0, \theta, n) = y_n$ where $y_n$ is defined recursively by $y_0 = z_0$ and

$$y_{n+1} = y_n - g(y_n, \theta). \tag{166}$$

Finally, we define $\Theta_f(z_0, n) = \arg\min_\theta \ell(\phi_f(z_0, \theta, n))$ and $\Theta_g(z_0, n) = \arg\min_\theta \ell(\phi_g(z_0, \theta, n))$. These are the solutions of the problem (163) where we use the approximation $z^*(\theta) = \phi_f(z_0, \theta, n)$ (resp. $z^*(\theta) = \phi_g(z_0, \theta, n)$).

The parameter $\Theta_f$ corresponds to the approximate solution to the implicit deep learning found in practice with the true non-linear function $f$. The parameter $\Theta_g$ corresponds to the setting studied in Section 3. We seek to answer:

**Question** How far are the true parameters $\Theta_f$ from those found with the linear approximation $\Theta_g$?

## I.1 CONTROL OF THE DISTANCE OF THE ITERATES OF THE POWER METHOD

In order to control this distance, we begin by controlling the distance between the iterates of the power method with $f$ and $g$. We have:

**Lemma I.3** (Crude control of the norm of $z_n$). *Under Assumption I.1 and Assumption I.2, we have that if $\|z_0 - z^*\| \leq \frac{1-\lambda}{2\alpha}$ and $\|\theta - \theta^*\| \leq \frac{\sqrt{\|B\|_2^2 + \frac{(1-\lambda)^2\beta}{\alpha}} - \|B\|_2}{2\beta}$, then for all $n$, $\|z_n - z^*\| \leq \frac{1-\lambda}{2\alpha}$; in other words, $\|\phi_f(z_0, \theta, n) - z^*\| \leq \frac{1-\lambda}{2\alpha}$.*

*Proof.* First, note that the condition on $\theta$ implies:

$$\|B(\theta - \theta^*)\| + \beta\|\theta - \theta^*\|^2 \leq \|B\|_2\|\theta - \theta^*\| + \beta\|\theta - \theta^*\|^2 \tag{167}$$

$$\leq \frac{(1-\lambda)^2}{4\alpha}, \tag{168}$$

since the upper bound of $\|\theta - \theta^*\|$ is the positive root of the equation $\|B\|_2 x + \beta x^2 = \frac{(1-\lambda)^2}{4\alpha}$.
Next, we can rewrite the recursion on $z_n$ as

$$z_{n+1} = z_n - A(z_n - z^*) - B(\theta - \theta^*) - R(z_n, \theta).$$

Subtracting $z^*$, taking norms, applying the triangular inequality, and using Assumption I.1 and Assumption I.2 leads to

$$\|z_{n+1} - z^*\| \leq \lambda\|z_n - z^*\| + \alpha\|z_n - z^*\|^2 + \|B(\theta - \theta^*)\| + \beta\|\theta - \theta^*\|^2 \tag{169}$$

and using the bound (167) we find the recursive inequality

$$\|z_{n+1} - z^*\| \leq \lambda\|z_n - z^*\| + \alpha\|z_n - z^*\|^2 + \frac{(1-\lambda)^2}{4\alpha}.$$

Hence, if $\|z_n - z^*\| \leq \frac{1-\lambda}{2\alpha}$, we find that

$$\|z_{n+1} - z^*\| \leq \lambda\frac{1-\lambda}{2\alpha} + \alpha\frac{(1-\lambda)^2}{4\alpha^2} + \frac{(1-\lambda)^2}{4\alpha} = \frac{1-\lambda}{2\alpha},$$

which demonstrates the result by induction. $\qquad\square$

We now use the previous bound to "remove" the challenging quadratic term in the recursive inequality followed by $\|z_n - z^*\|$ and then obtain a far better control.

**Lemma I.4** (Tighter control of the norm of $z_n$). *We assume the same hypotheses as in Lemma I.3. We also assume that $\|\theta - \theta^*\| \leq \frac{\|B\|_2}{\beta}$. Then, for all $n$, we have $\|z_n - z^*\| \leq \|z_0 - z^*\| + \frac{4\|B\|_2}{1-\lambda}\|\theta - \theta^*\|$.*

*Proof.* The new assumption on $\theta$ and the bound from the previous lemma plugged in Eq. (169) give the simple recursion

$$\|z_{n+1} - z^*\| \leq \frac{1+\lambda}{2}\|z_n - z^*\| + 2\|B\|_2\|\theta - \theta^*\|$$

Unrolling it leads to

$$\|z_n - z^*\| \le \left(\frac{1+\lambda}{2}\right)^n \|z_0 - z^*\| + \frac{1 - \left(\frac{1+\lambda}{2}\right)^n}{1 - \frac{1+\lambda}{2}} 2\|B\|_2 \|\theta - \theta^*\|$$

Crudely upper-bounding $\left(\frac{1+\lambda}{2}\right)^n \le 1$ and $1 - \left(\frac{1+\lambda}{2}\right)^n \le 1$ leads to the advertized bound. $\qquad \square$

This bound shows that the distance between $z_n$ and the solution $z^*$ is linear in the distance between $(z_0, \theta)$ and $(z^*, \theta^*)$; which is quite expected: indeed, starting from $(z_0, \theta) = (z^*, \theta^*)$ in the procedure leads to $z_n = z^*$ for all $n$; it is only natural that $z_n$ does not depart too much from $z^*$ as $(z_0, \theta)$ moves away from $(z^*, \theta^*)$.

We now turn to controlling the distance between the iterates produced by $f$ and those produced by $g$.

**Lemma I.5** (Control of the distance between $z_n$ and $y_n$). *We assume the same hypotheses as in Lemma I.4. Define the constants $\mu = \frac{2\alpha}{(1-\lambda)}$ and $\nu = \left(\frac{\beta}{1-\lambda} + \frac{32\alpha\|B\|_2^2}{(1-\lambda)^3}\right)$. Then, for all $n$, it holds*

$$\|z_n - y_n\| \le \mu\|z_0 - z^*\|^2 + \nu\|\theta - \theta^*\|^2.$$

*In other words, $\|\phi_f(z_0, \theta, n) - \phi_g(z_0, \theta, n)\| \le \mu\|z_0 - z^*\|^2 + \nu\|\theta - \theta^*\|^2$.*

*Proof.* We define $r_n = z_n - y_n$. This sequence verifies the recursion

$$r_{n+1} = r_n - Ar_n + R(z_n, \theta)$$

starting from $r_0 = 0$. Taking norms and upper-bounding, we get

$$\|r_{n+1}\| \le \lambda\|r_n\| + \alpha\|z_n - z^*\|^2 + \beta\|\theta - \theta^*\|^2.$$

Squaring the bound of Lemma I.4 and using $(a + b)^2 \le 2a^2 + 2b^2$ gives $\|z_n - z^*\|^2 \le 2\|z_0 - z^*\|^2 + \frac{32\|B\|_2^2}{(1-\lambda)^2}\|\theta - \theta^*\|^2$, so we get the recursion

$$\|r_{n+1}\| \le \lambda\|r_n\| + 2\alpha\|z_0 - z^*\|^2 + \left(\beta + \frac{32\alpha\|B\|_2^2}{(1-\lambda)^2}\right)\|\theta - \theta^*\|^2.$$

Unrolling this recursion gives

$$\|r_n\| \le \frac{2\alpha}{1-\lambda}\|z_0 - z^*\|^2 + \left(\frac{\beta}{1-\lambda} + \frac{32\alpha\|B\|_2^2}{(1-\lambda)^3}\right)\|\theta - \theta^*\|^2.$$

$\qquad \square$

This result is central for the rest of our analysis: it shows that the distance between the iterates of the two power methods is *quadratic* in the distance of $(z_0, \theta)$ to $(z^*, \theta^*)$.

### I.2 CONTROL OF THE DISTANCE OF THE OPTIMAL PARAMETERS

The previous analysis (Lemma I.5) demonstrates that if $\theta$ is close to $\theta^*$ and $z^0$ is close to $z^*$, then the iterates of power method using $f$ and those of the power method using $g$, the linearization of $f$, are extremely close. This, in turn, implies that the optimal parameters to the corresponding bilevel problems are close.

**Assumption I.6.** The function $\ell$ has $z^*$ as a unique minimizer.

**Assumption I.7.** The matrix $B$ is invertible.

Under this assumption, we can derive the expression of the optimal parameters $\theta$ found with the power method applied on $g$

**Lemma I.8** (Optimal parameters for $g$). *Under Assumption I.2, Assumption I.6, and Assumption I.7, we have that*

$$\Theta_g(z_0, n) := \arg\min_\theta \ell(\phi_g(z_0, \theta, n)) = \theta^* + B^{-1}(I - (I - A)^n)^{-1}A(I - A)^n(z_0 - z^*)$$

*Proof.* Unrolling the affine recursion for $g$ yields

$$\phi_g(z_0, \theta, n) = z^* + (I - A)^n(z_0 - z^*) - (I - (I - A)^n)A^{-1}B(\theta - \theta^*)$$

By assumption, the function $\ell(\phi_g(z_0, \theta, n))$ is minimized for when $\phi_g(z_0, \theta, n) = z^*$, which gives the advertized result. $\qquad \square$

In order to control the error induced by using $f$, we need the following technical lemma:

**Lemma I.9.** *Let $\psi : \mathbb{R}^d \to \mathbb{R}^d$ a continuous function such that for some $\delta, \gamma > 0$ with $\delta\gamma \leq \frac{1}{4}$, we have*

$$\forall u \in \mathbb{R}^d, \ \ \|\psi(u)\| \leq \delta + \gamma\|u\|^2.$$

*Then, there exists $u^*$ such that $\psi(u^*) = u^*$ with $\|u^*\| \leq 2\delta$.*

*Proof.* Let $u$ such that $\|u\| \leq 2\delta$. Then, $\|\psi(u)\| \leq \delta + \gamma\|u\|^2 \leq \delta + 4\gamma\delta^2 \leq 2\delta$. In other words, the ball centered in 0 of radius $2\delta$ is stable by $\psi$. As a consequence, Brouwer's fixed point theorem implies that $\psi$ has a fixed point in that ball. $\square$

We are now ready for the main result:

**Theorem I.10.** *We assume the same hypotheses as in Lemma I.5 and Lemma I.8. Define the constant $\Delta_0 = \|B^{-1}\|_2(\mu + 2\nu\|B^{-1}\|_2)$, and assume that $z_0$ is such that $\|z_0 - z^*\| \leq (8\|B^{-1}\|_2\nu\Delta_0)^{-\frac{1}{2}}$. Then, the problem $\min_\theta \ell(\phi_f(z_0, \theta, n))$ has a solution $\Theta_f(z_0, n)$ such that*

$$\|\Theta_f(z_0, n) - \Theta_g(z_0, n)\| \leq 2\Delta_0\|z_0 - z^*\|^2.$$

*Proof.* We let $R'(z_0, \theta, n) = \phi_f(z_0, \theta, n) - \phi_g(z_0, \theta, n)$ the distance between the iterates of the two power methods, which is upper-bounded following Lemma I.5. A minimizer of $\ell(\phi_f(z_0, \theta, n))$ is found by solving $\phi_f(z_0, \theta, n) = z^*$, i.e. by finding a solution to the equation

$$\phi_g(z_0, \theta, n) = z^* + R'(z_0, \theta, n).$$

Using the previous closed-form expression for $\phi_g$, this is equivalent to finding $\theta$ such that

$$\theta = \Theta_g(z_0, n) - B^{-1}R'(z_0, \theta, n)$$

We define the new variable $u = \theta - \Theta_g(z_0, n)$, and define $\psi(u) = -B^{-1}R'(z_0, u + \Theta_g(z_0, n), n)$. The solutions of the previous equations are exactly the fixed points of $\psi$.

We have, using Lemma I.5:

$$\|\psi(u)\| \leq \|B^{-1}\|_2(\mu\|z_0 - z^*\|^2 + \nu\|u + \Theta_g(z_0, n) - \theta^*\|^2 \tag{170}$$
$$\leq \|B^{-1}\|_2(\mu\|z_0 - z^*\|^2 + \nu\|u + B^{-1}(I - (I - A)^n)^{-1}A(I - A)^n(z_0 - z^*)\|^2 \tag{171}$$
$$\leq \delta + \gamma\|u\|^2 \tag{172}$$

with $\delta = \|B^{-1}\|_2(\mu + 2\nu\|B^{-1}(I - (I - A)^n)^{-1}A(I - A)^n\|_2)\|z_0 - z^*\|^2$ and $\gamma = 2\|B^{-1}\|_2\nu$. Using $\|(I - (I - A)^n)^{-1}A(I - A)^n\|_2 \leq 1$ further simplifies the formula of $\delta$ to $\delta = \Delta_0\|z_0 - z^*\|^2$ Hence, following the previous lemma, if $2\|B^{-1}\|_2\nu \times \Delta_0\|z_0 - z^*\|^2 \leq \frac{1}{4}$, $\psi$ has a fixed point of norm less than $2\delta$, in other words there is a solution $\Theta_f(z_0, n)$ to $\min_\theta \ell(\phi_f(z_0, \theta, n))$ such that $\|\Theta_f(z_0, n) - \Theta_g(z_0, n)\| \leq 2\Delta_0\|z_0 - z^*\|^2$ $\square$

This theorem shows that the solutions of the non-linear practical bilevel problem and those of the linear approximation are very close when the initialization is close to the solution.

### I.3  I2O FOR NON-LINEAR IMPLICIT MODELS

We are now ready to bound the quantity $D_f(N, \Delta N) = \ell(z_{N+\Delta N}(\theta^*_{f,N})) - \ell(z_N(\theta^*_{f,N}))$, where $\theta^*_{f,N} = \Theta_f(z_0, N)$ as defined in the previous section and $z_N$ is obtained with a power method on $f$. Our affine theory from Theorem 1 allows us to control $D_g(N, \Delta N) = \ell(y_{N+\Delta N}(\theta^*_{g,N})) - \ell(y_N(\theta^*_{g,N}))$ where $\theta^*_{g,N} = \Theta_g(z_0, N)$ and $y_N$ is obtained with a power method on $g$.
We have the following control:

**Theorem I.11.** *We assume that the cost function $\ell$ is L-Lipschitz, and the hypotheses of Theorem I.10. Then:*

$$|D_f(N, \Delta N) - D_g(N, \Delta N)| \leq \kappa\|z_0 - z^*\|^2,$$

*with $\kappa = 2L(\frac{2\|B\|_2\Delta_0}{1-\lambda} + \mu + \nu(\Delta_0(\|B^{-1}\|_2\nu)^{-1} + 2\|B^{-1}\|_2^2))$*

*Proof.* We control:

$$|\ell(y_{N+\Delta N}(\theta^*_{g,N})) - \ell(z_{N+\Delta N}(\theta^*_{f,N}))| \le L\|y_{N+\Delta N}(\theta^*_{g,N}) - z_{N+\Delta N}(\theta^*_{f,N})\|$$
$$\le L(\|y_{N+\Delta N}(\theta^*_{g,N}) - y_{N+\Delta N}(\theta^*_{f,N})\| + \|y_{N+\Delta N}(\theta^*_{f,N}) - z_{N+\Delta N}(\theta^*_{f,N})\|)$$

The first term is controlled by regularity of $y_{N+\Delta N}$:

$$\|y_{N+\Delta N}(\theta^*_{g,N}) - y_{N+\Delta N}(\theta^*_{f,N})\| \le \frac{\|B\|_2}{1-\lambda}\|\theta^*_{g,N} - \theta^*_{f,N}\|$$

and then using Theorem I.10, we get

$$\|y_{N+\Delta N}(\theta^*_{g,N}) - y_{N+\Delta N}(\theta^*_{f,N})\| \le \frac{2\|B\|_2\Delta_0}{1-\lambda}\|z_0 - z^*\|^2.$$

The second term is controlled using Lemma I.5, which gives

$$\|y_{N+\Delta N}(\theta^*_{f,N}) - z_{N+\Delta N}(\theta^*_{f,N})\| \le \mu\|z_0 - z^*\|^2 + \nu\|\theta^*_{f,N} - \theta^*\|^2.$$

Next, we bound crudely:

$$\|\theta^*_{f,N} - \theta^*\|^2 \le 2\|\theta^*_{f,N} - \theta^*_{g,N}\|^2 + 2\|\theta^*_{g,N} - \theta^*\|^2.$$

The first term is upper bounded using Theorem I.10:

$$\|\theta^*_{f,N} - \theta^*_{g,N}\|^2 \le 4\Delta_0^2\|z_0 - z^*\|^4 \le \Delta_0(2\|B^{-1}\|_2\nu)^{-1}\|z_0 - z^*\|^2$$

while the second is upper-bounded by Lemma I.8: $\|\theta^*_{g,N} - \theta^*\|^2 \le \|B^{-1}\|_2^2\|z_0 - z^*\|^2$, giving:

$$\|\theta^*_{f,N} - \theta^*\|^2 \le (\Delta_0(\|B^{-1}\|_2\nu)^{-1} + 2\|B^{-1}\|_2^2)\|z_0 - z^*\|^2.$$

We then get

$$\|y_{N+\Delta N}(\theta^*_{f,N}) - z_{N+\Delta N}(\theta^*_{f,N})\| \le (\mu + \nu(\Delta_0(\|B^{-1}\|_2\nu)^{-1} + 2\|B^{-1}\|_2^2))\|z_0 - z^*\|^2.$$

Overall, we obtain

$$|\ell(y_{N+\Delta N}(\theta^*_{g,N})) - \ell(z_{N+\Delta N}(\theta^*_{f,N}))| \le L(\frac{2\|B\|_2\Delta_0}{1-\lambda} + \mu + \nu(\Delta_0(\|B^{-1}\|_2\nu)^{-1} + 2\|B^{-1}\|_2^2))\|z_0 - z^*\|^2.$$

Finally, we get the advertised result by doing

$$|D_f(N, \Delta N) - D_g(N, \Delta N)| \le |\ell(y_{N+\Delta N}(\theta^*_{g,N})) - \ell(z_{N+\Delta N}(\theta^*_{f,N}))| + |\ell(y_N(\theta^*_{g,N})) - \ell(z_N(\theta^*_{f,N}))|$$
$$\le 2L(\frac{2\|B\|_2\Delta_0}{1-\lambda} + \mu + \nu(\Delta_0(\|B^{-1}\|_2\nu)^{-1} + 2\|B^{-1}\|_2^2))\|z_0 - z^*\|^2.$$

$\square$

This result shows that the practical gap $\Delta_f(N, \Delta N)$ is very well approximated by $\Delta_g(N, \Delta N)$, the gap of the linear approximation of $f$, provided that the initiliazation of the model $z_0$ is not too far from the solutions $z^*$.

