# OpenReview forum: "Test like you Train in Implicit Deep Learning"
_ICLR.cc/2024/Conference — Submitted to ICLR 2024_

### Official Review · Reviewer_uPNB · 2023-11-01

**Soundness:** 2 fair
**Presentation:** 2 fair
**Contribution:** 2 fair
**Rating:** 5
**Confidence:** 3

**Summary:**

The paper investigates implicit deep learning, particularly in meta-learning and Deep Equilibrium Networks (DEQs). A common belief is that increasing the number of iterative solutions (inner iterations) during inference improves performance. This study challenges that notion, providing a theoretical analysis that highlights overparametrization as a crucial factor. The findings reveal that overparametrized networks, like DEQs, don't benefit from extra iterations at inference, while meta-learning does.

**Strengths:**

The topic is interesting, and the paper clearly states the motivation, system model, and assumptions.

**Weaknesses:**

1. It is hard to interpret the meaning of main theorems. For example, Eq. (9) in Theorem 1 contains orthogonal projections and $E_N$ while their values are unknown.

2. The tightness of the bounds shown in these theorems is not discussed. It will be beneficial to use a figure to compare the actual value and the analytical bound.

**Questions:**

The major questions I have are about $D(N,\Delta N)$, i.e., the change of the training loss after changing number of inner iterations by $\Delta N$ for a fixed learned $\theta^{\*,N}$. First, this definition seems problematic because when $N$ changes, $\theta^{\*,N}$ should change. In other words, we should not fix $\theta^{\*,N}$. Second, this is on training loss, not the test loss. The relationship between training loss and test loss is not trivial, especially when overparameterized. In the paper, the authors claim that "This quantity is a proxy for the increase in test loss, provided we have access to enough training data", which I doubt since the meaning of "enough training data" is very vague (or even contradictory) in the context of overparameterization.

---

> ### Author Response · Authors · 2023-11-18
>
> We would like to thank the reviewer for their review.
> We tackled the question of the discrepancy between train and test in the common response (*C1*).
> We will answer each point subsequently.
> - **Explanation of theorem 1**: Theorem 1 states that it is possible to improve the loss by changing the number of iterations only up to a “point”, and gives a quantitative result for this “point” (i.e. the lower bound). We even show in the subsequent Corollary 1, that in the most extreme case it is not possible to improve the loss.
> We will add this clarification after theorem 1 in the camera-ready version of our paper.
> We recall that all notations are defined within theorem 1 or its assumptions.
> - **Tightness of the bound**: The bound is tight and we will mention it in the camera-ready version of our paper. It comes from the fact that only triangular inequalities were used in the proof. Fig 1. (left) shows the tightness under the assumptions of Corollary 1.
> - **Adapting $\theta^{\star, N}$**: Indeed, we agree that this point is not intuitive, but two aspects justify such a framing of the problem: 1. It is the current practice not to refit the model and use it with a different number of inner iterations as we found in all the state of the art implementations. This is therefore the mathematical equation behind what happens in practice. 2. It has been claimed in the literature that DEQs have the possibility to generalize beyond the $N$ for which it was trained. We wanted to show that, indeed, it is important in the current state to use DEQs with the $N$ for which it was trained.

---

> > ### Comment · Reviewer_uPNB · 2023-11-23
> >
> > Thanks for the explanation. I keep my original rating.

---

### Official Review · Reviewer_YcSM · 2023-11-01

**Soundness:** 3 good
**Presentation:** 2 fair
**Contribution:** 2 fair
**Rating:** 6
**Confidence:** 2

**Summary:**

This paper focuses on the inner iterations overfitting problem of overparameterized models in implicit deep learning, and provides theoretical results in a simplified affine setting to show that increasing the number of iterations at test time cannot improve
performance for overparametrized models. Two typical implicit deep learning methods, DEQ and iMAML, are considered in the paper. Experiments on diverse tasks verifies the theorem on both DEQ and iMAML.

**Strengths:**

1. The paper is well written and easy to follow.
2. The definition and analysis of inner iterations overfitting problem is novel and will be helpful for future researches on implicit deep learning.

**Weaknesses:**

Experiments can be further improved. Fore example,
- it only considers the case where $N$ is fixed while $\Delta N$ varies (e.g., Figure 2 & 3). Does the conclusion hold for other choices of $N$?
- Theorem 1 is validated on a small scale dataset. It will be helpful to validate it on real dataset.

**Questions:**

See the section of Weaknesses

---

> ### Author Response · Authors · 2023-11-18
>
> We would like to thank the reviewer for their review.
> We will answer each point subsequently.
> - **Varying $N$**: In each experiment, $N$ is different, in other words we don’t use the same number of inner iterations for different settings. The idea here was to reuse pre-trained DEQs, which generally have very fine hyperparameter tuning to maximize the performance on the task and for a specific number of inner iterations. Each $N$ is specified in Table 1. in the Appendix.
> Using different $N$ would mean retraining all of these models ourselves, which is beyond our computational budget.
> We also wanted to make sure to uncover this phenomenon for models that were trained by others in order to strengthen our findings.
> - **Validation of Theorem 1**: we tried to verify Corollary 2 on a more real dataset in Figure 4. Figure 1. was indeed meant as an illustration rather than a validation of the theorem, and we believe that our experiments on diverse real-world dataset demonstrate the applicability of our findings.

---

### Official Review · Reviewer_UkB2 · 2023-11-05

**Soundness:** 2 fair
**Presentation:** 3 good
**Contribution:** 2 fair
**Rating:** 3
**Confidence:** 3

**Summary:**

This paper studies when increasing the number of fixed-point iterations in implicit deep learning during test time improves performance. In implicit deep learning, the network must _implicitly_ minimize the objective $\ell(z(\theta))$. Specifically, the network parameters $\theta$ control some intermediary output $z(\theta)$ which tends to be the solution of some inner rootfinding problem $f(z, \theta, D_{train}) = 0$. The solution $z(\theta)$ is often identified through some $N$ fixed-point updates during training, then during test time, the solution is updated using test data using more fixed-point updates. This helps in certain applications like meta-learning and not others like DEQ.

The paper attempts to prove that this difference (helping in certain cases and not others) can be explained by the degree $\theta$ is overparameterized for the inner problem. They prove lower bounds for how much the training loss can change when changing the number of fixed-point iterations for an affine inner problem.

**Strengths:**

- The paper is generally clear and easy to follow.
- Excluding a couple minor things (bullet points below), the proofs in Appendix Section A for the main theorems in the main body seem to be correct.
    - Between Eq. 22 and 23, it says “Therefore $z_{N+1}$ is also in the range of $K^\top$, should say $z_{N+1} - z_0$.
    - Eq 30 has some problems with the signs, although the final form Eq 31 seems to be correct.
- The empirical results in Figures 2/3 seem to support their hypothesis that overparameterization leads to less improvements with more inner updates.

**Weaknesses:**

- As already pointed out by the authors, there is a major inconsistency between what the paper tries to prove and what is actually proven. During inference, the inner optimization is conducted on the _test data_. However, the paper’s results are only showing how the _training_ loss can fluctuate with the number of inner optimization updates. The problem with this relaxation is because it interferes directly with the paper’s core result, that network overparametrization is why more inner optimization steps do not help. One can imagine in the overparametrized setting, the test loss/inner problem could be significantly different from the training loss/inner problem, causing an analysis on just the training loss to fall apart. I may be misunderstanding the paper however, and so I vote for a low score with low confidence.

- I think the authors can be more precise about what they mean by overparametrization. Formally, they do define it as $d_x < d_\theta$ in Corollary 1, but it might be good to clarify how $d_x$ scales for the different applications (DEQ, meta-learning) in the main body. Is this definition related to the classic usage of the term “overparametrization” as the number of parameters being larger than the number of training data in linear models? And the word overparametrization is a bit thrown around loosely in the paper.

- Figure 4 plots $D(N, \Delta N)$ of the training loss for different levels of “overparameterization” which they measure using the training loss (lower training loss → more expressive model). They conclude from the experiment that “The lower the training loss, the higher the $D(N, \Delta N)$”. This comparison seems a bit unfair and requires normalizing $D(N, \Delta N)$ with respect to the training loss. For a loss lower bounded by 0 (like most losses), if the training loss is already small, the lowest possible improvement with more fixed point iterations $D(N, \Delta N)$ obviously gets smaller.


- There’s some minor spacing issue of Figure 2

**Questions:**

- What is “convergence” plotted in FIgure 2?
- I am not too familiar with meta-learning literature and am a bit confused about doing the fixed-point iterations during inference, specifically for meta-learning. As implied by Equation 5, it seems to require access to test labels..Is the inner optimization during inference conducted on the test data using labels?

---

> ### Author Response · Authors · 2023-11-18
>
> We would like to thank the reviewer for engaging in the review process benevolently.
> We tackled the question of the discrepancy between train and test in the common response (*C1*).
> We will answer all other raised points subsequently.
>
> - **Typos in the proofs**: Thank you for spotting these, we correct them in our revised version.
> - **Clarification of overparametrization**:
> We take a clear definition of over-parameterization, which is that we can learn any vector in our latent space; this should be connected to the number of training points. This clear definition does not necessarily match the usual definition of #params > #samples, but it has the same intuitive flavour of having a large enough number of parameters to be able to model a wide class of functions.
> Then, there's the practical definition of overparameterization, in a practical network that does not necessarily have the same architecture as the one studied.
> - **Fig 4: fairness of comparison across losses**: We think this is a very good point, and have reworked Fig 4. to include this normalization. It should be noted that with the normalization the result is less striking for MAML, but we still obtain a negative correlation score of $-0.4$ with a p-value of $0.03$ (i.e. statistically significant) without rerunning the experiment. For iMAML the result is still very obvious with a negative correlation score of $-0.6$ and a p-value of $7.10^{-4}$.
> - **Fig 2**: spacing thanks for noticing, we will correct this in the camera-ready version of our work.
> - **Convergence**: In fig 2., convergence is the norm of the error on the fixed point, i.e. in our notations $\|z_{N+\Delta N}(\theta^{\star, N}) - f(z_{N+\Delta N}(\theta^{\star, N}), \theta^{\star, N})\|$. We clarify this in our revised version in the caption of Fig 2.
> - **Meta-learning with validation**: Indeed, in order to solve the meta-learning problem, we need to use a set of pairs of datasets: each pair has a training set and a validation set. During training, meta-learning use both the training and validation set in order to learn an initialization (or anchor weights) that when fitted on the training data allows to generalize well when evaluated on the validation data.
> When testing with weights $\theta^{(\text{meta})}$, we use other pairs of training and validation datasets, as done by Finn et al. 2017 and Rajeswaran et al. 2019.

---

### Author Response · Authors · 2023-11-18

We thank the reviewers for engaging in the review process and hope that the rebuttal period will be fruitful.
The reviewers praised our work for being “well written and easy to follow.” (YcSM) and found our “empirical results [...] support our hypotheses” (UkB2).
One of the main points raised by the reviewers was the discrepancy between train and test. We give here a common answer:
**C1: Discrepancy train/test**: Indeed, the discrepancy between train and test data is a major point that we highlighted (see page 6.). However, in the case of DEQs, we also provided a generalization result as stated on page 6 “We provide a generalization result in Corollary D.1 (in Appendix D) for DEQs stacked sample-wise with some additional smoothness assumptions.”. This result shows that if we have access to enough data and the functions $K_\text{in}(x_i), B(x_i), \ldots$ are sufficiently smooth, then theorem 1 holds on average for test data. This is equivalent to the kinds of results one usually gets when applying the Probably Approximately Correct (PAC) framework, which is generally used to discuss the generalization of machine learning approaches (our analysis is inspired by such literature) for example Mohri et al. 2018.
The takeaway is that we proved that given sufficient data (which incidentally is also a necessary assumption for a model to generalize) our analysis holds in the test case. This is further validated by our experiments.
We further highlight this aspect in our revised version by saying in the introduction “We also characterize the implications of this result for non-affine f in Theorem I.10, and for the generalization setting in Corollary D.1.” and also mention it in the empirical part.
To be more specific on the evolution of overparametrization in the case of increasingly large training datasets, one can see that it’s possible to remain overparametrized even in this case, i.e. have $\mathcal{B}\_N(\mathcal{D}\_\text{train}) = 0$ (defined in eq 132 in Appendix). This typically happens in the trivial case where all the functions $U, B, K_\text{in}, \ldots$ are constant. More generally, the bound does not diverge even when increasing the number of data points if the data support is compact.
We added a short paragraph covering this in the generalization appendix.

---

### Author Response · Authors · 2023-11-22
**End of discussion period**

For this last day of the discussion period we would like to summarize our responses. Mainly we have answered the point raised by 2 reviewers on the discrepancy between train and test, highlighting our current results (both theoretical and empirical) on this point. We also proceeded to some clarifications regarding our results, and modified a Figure (fig 4) to better illustrate our point.
We think that all the points raised by the reviewers have been addressed satisfactorily and we would be grateful to the reviewers should they consider raising their score to accept if they think similarly.
Thanks again for your commitment to the review.

---

### Meta-Review · Area_Chair_LRQv · 2023-12-09

**Metareview:**

This paper studies the role of the number of inner iterations in implicit deep learning during test time on the performance. The paper attributes this role to the degree of overparameterized for the inner problem. They prove lower bounds for how much the training loss can change when changing the number of fixed-point iterations for a simple affine inner problem. The theory is validated on DEQs and meta-learning problems.

The reviewers found the paper clear to read, technically correct, and having good empirical support for their theory. However, multiple reviewers pointed out a major issue which is that the main theoretical result considers the training loss rather than the test loss, which is inconsistent with the main message of the paper. The authors pointed out a generalization result in Appendix D as a resolution of this issue, but that is based on a PAC-style argument and it isn't clear to the reviewers and the area chair how this applies to the setting considered, especially with overparameterization. The authors are encouraged to prove an explicit theorem on test data in future revisions.

**Justification For Why Not Higher Score:**

There's a disconnect between the main message of the paper and the theorem actually proved.

**Justification For Why Not Lower Score:**

N/A

---

### Decision · Program_Chairs · 2024-01-16

Reject